
# Reconstructing the graviton

Alfio Bonanno[1,2], Tobias Denz[3], Jan M. Pawlowski[3,4] and Manuel Reichert[5]

1 INAF, Osservatorio Astrofisico di Catania, via S. Sofia 78, 95123 Catania, Italy
2 INFN, Sezione di Catania, via S. Sofia 64, 95123 Catania, Italy
3 Institut für Theoretische Physik, Universität Heidelberg,
Philosophenweg 16, 69120 Heidelberg, Germany
4 ExtreMe Matter Institute EMMI, GSI Helmholtzzentrum für Schwerionenforschung mbH,
Planckstr. 1, 64291 Darmstadt, Germany
5 Department of Physics and Astronomy, University of Sussex, Brighton, BN1 9QH, U.K.

## Abstract

We reconstruct the Lorentzian graviton propagator in asymptotically safe quantum gravity from Euclidean data. The reconstruction is applied to both the dynamical fluctuation graviton and the background graviton propagator. We prove that the spectral function of the latter necessarily has negative parts similar to, and for the same reasons, as the gluon spectral function. In turn, the spectral function of the dynamical graviton is positive. We argue that the latter enters cross sections and other observables in asymptotically safe quantum gravity. Hence, its positivity may hint at the unitarity of asymptotically safe quantum gravity.

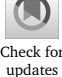

# 1 Introduction

In the past two decades, asymptotically safe (AS) gravity has emerged as an interesting and solid contender for a quantum theory of gravity. By now a lot of non-trivial evidence has been collected for the existence of AS gravity as an ultraviolet (UV) complete quantum field theory. Most of the investigations have been done with the functional renormalisation group (fRG), for a recent overview see [1]. For reviews on AS gravity including its application to high energy physics see [1–14]. However, there are some pivotal challenges yet to be resolved, see [13]. A very prominent one is the setup of a non-perturbative Lorentzian signature approach: most investigations so far have been done within Euclidean quantum gravity. The Wick rotation to a Lorentzian version is one of the remaining challenges yet to be met. For first steps towards Lorentzian flows see, e.g., [15–24], for work in related quantum gravity approaches see, e.g., [25–31], and for discussions of ghosts and the Ostrogradsky instability see, e.g., [32–36]. The proper definition of the Wick rotation and the interpretation of the spectral properties of Lorentzian correlation functions also touch upon the question of unitarity of the approach.

A first, but important, step in this direction is done by the reconstruction of spectral functions from their Euclidean counterparts. While being short of a full resolution of the challenges mentioned about, it provides non-trivial insight into the possible complex structure of AS gravity. In the present work, we apply reconstruction methods, already used successfully in non-Abelian gauge theories [37], to the fundamental correlation function of AS gravity, the graviton two-point function or propagator. In (non-Abelian) gauge theories, as in gravity, one has to face the fact that the gauge field is not an on-shell physical field. Hence, the standard derivation of

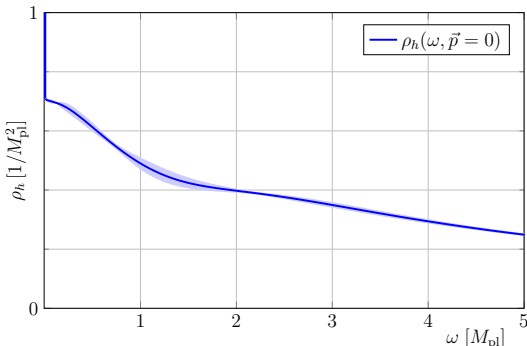

(a) Spectral function of the fluctuation graviton.

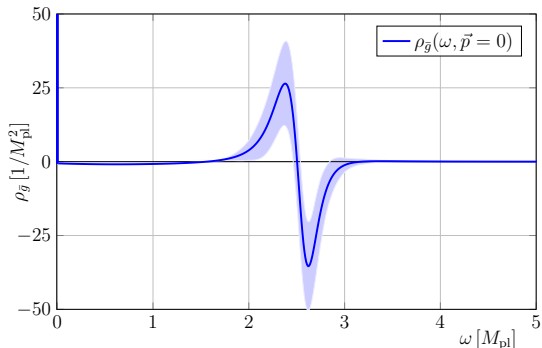

(b) Spectral function of the background graviton.

Figure 1: Spectral functions of the fluctuation and background graviton. The shaded areas constitute the estimated error of the reconstruction. The fluctuation graviton spectral function is strictly positive, which may be important for the unitarity of asymptotically safe gravity. In turn, we show that the background graviton spectral function has a vanishing spectral weight and hence positive and negative parts. For more details see Sec. 5.2 and Sec. 5.3.

the Källén–Lehmann spectral representation fails, and the spectral function of the gauge field features non-positive parts if it exists. This property follows already in the perturbative high-momentum regime of the gluon from the Oehme-Zimmermann super-convergence property. Consequently, non-positive parts of spectral functions have but nothing to do with strongly-correlated physics such as confinement, and, even more importantly, do not signal the failure of unitarity of the theory. In turn, they also should not be taken lightly.

Similar properties may be present in AS quantum gravity. For a first discussion see [38], for related recent work see [39–41]. In the present work, we formally show that the spectral function of the background graviton has non-positive parts. We also compute the spectral function of the dynamical fluctuation graviton, which turns out to be positive. As the derivation of these spectral functions requires many steps and intermediate results, we present the final spectral functions already in Fig. 1. The knowledge of the structures visible there certainly allows to better follow some of the technical steps, and also understand their origin. In particular, the positivity of the spectral function of the fluctuation graviton in Fig. 1 is a non-trivial and very exciting result, as it is the fluctuation graviton which propagates in the diagrams of cross-sections and other potential observables. While the results leave much to be explained, the present work is a first rather non-trivial step towards a discussion of unitarity in AS gravity.

This work is structured as follows: In Sec. 2, we give a brief overview of the setup of our approach to AS quantum gravity as well as the fRG approach used for the computation of Euclidean correlation functions. In Sec. 3, we introduce the Källén–Lehmann spectral representation and discuss the properties of the graviton spectral function. We also prove that the spectral function of the background graviton has negative parts. In Sec. 4, we present our computation of Euclidean correlation functions, mainly based on results in previous works. In Sec. 5, we introduce our reconstruction framework, compute and discuss our results on graviton spectral functions. A summary and outlook of our findings can be found in Sec. 6.

## 2 Asymptotically safe gravity

Asymptotically safe quantum gravity [42,43] is a viable and minimal UV closure of fundamental physics. It is minimal in the sense that it only relies on standard quantum field theory, and its viability has been furthered by many results in the past two decades following the seminal paper [44] within the fRG-approach, see [1–14]. Most of these fRG-based investigations are done within a Euclidean setting, also commonly used for other non-perturbative investigations, being short of numerically accessible approaches with Lorentzian signature.

The extraction of (timelike) physics from Euclidean correlation function is already intricate and challenging within standard high energy physics, and in particular for strongly correlated physics such as infrared (IR) QCD. However, in quantum gravity, a further, even conceptual, challenge is the very definition of a Wick rotation.

While chiefly important, we will not touch upon this crucial subject here, and simply assume the existence of a standard Wick rotation at least for backgrounds close to flat ones. This allows us to use standard reconstruction techniques for the computation of spectral properties of correlation functions in quantum gravity from their Euclidean counterparts. Hence, below we introduce the Euclidean approach to AS gravity.

### 2.1 Euclidean quantum gravity and the fRG

In the present work, we utilise results for momentum-dependent correlation function in a Euclidean flat background based on [45] within the fluctuation approach. This approach has been set-up in [45–48] and used, e.g., for pure gravity investigations in [45–57], and for gravity-matter systems in [54–68]; for a recent review see [14]. Here we briefly describe the computation and approximation scheme, for more details see these works.

Central to the functional approach to asymptotic safety is the effective action $\Gamma$, the quantum analogue of the classical action. Importantly, the RG-approach to AS quantum gravity does not rely on a specific classical action, but on a non-trivial fixed-point action at the UV Reuter fixed point. In turn, the large-scale physics in the IR is well described by the Einstein-Hilbert action,

$$S_{\text{EH}}[g_{\mu\nu}] = \frac{1}{16\pi G_{\text{N}}} \int \mathrm{d}^4x \, \sqrt{g} \left( 2\Lambda - R(g_{\mu\nu}) \right), \tag{1}$$

with the (IR) classical Newton constant $G_{\text{N}}$ and the abbreviation $\sqrt{g} = \sqrt{\det g_{\mu\nu}(x)}$. The definition of a graviton propagator, a pivotal ingredient to the approach, requires a gauge fixing. A standard linear gauge fixing requires the definition of a background metric, which also serves as the expansion point of the effective action. We use a linear split for the full metric,

$$g_{\mu\nu} = \bar{g}_{\mu\nu} + \sqrt{16\pi G_{\text{N}}} \, h_{\mu\nu}, \tag{2}$$

where $h_{\mu\nu}$ is the dynamical fluctuation field with mass dimension 1. Importantly, the fluctuation field carries the quantum fluctuations. In the present work, we use the flat $O(4)$-symmetric Euclidean metric for $\bar{g}_{\mu\nu}$. The gauge fixing is done within this background, and we take a de-Donder type gauge fixing, see App. A.

This formulation introduces a separate dependence of the effective action on the background metric and the fluctuation fields, $\Gamma = \Gamma[\bar{g}_{\mu\nu}, \phi]$, where $\phi$ is the fluctuation multi-field including the ghosts,

$$\phi_i = (h_{\alpha\beta}, \bar{c}_\mu, c_\nu). \tag{3}$$

We consider a vertex expansion of the effective action about the given Euclidean background $\bar{g}$,

$$\Gamma[\bar{g}_{\mu\nu}, \phi] = \sum_{n=0}^{\infty} \frac{1}{n!} \prod_{l=1}^{n} \int \mathrm{d}^4x_l \, \sqrt{\det \bar{g}_{\mu\nu}(x_l)} \, \phi_{i_l}(x_l) \Gamma^{(\phi_{i_1}\cdots\phi_{i_n})}[\bar{g}_{\mu\nu}, 0](\mathbf{x}), \tag{4}$$

where $\Gamma^{(\phi_{i_1}\cdots\phi_{i_n})}$ are the $n$th derivatives of the effective action with respect to the fluctuation fields $\phi_{i_1}, ..., \phi_{i_n}$ and $\boldsymbol{x} = (x_1, \ldots, x_n)$. We shall also use the abbreviation $\Gamma^{(n)}$ for the sake of simplicity.

A suggestive choice for the background metric is a solution of the quantum equations of motion. There, background independence is regained and we expect the most rapid convergence of the vertex expansion, for a detailed discussion see e.g. [14]. Such an expansion is technically very challenging, and in the present work we consider an expansion about the flat Euclidean background, $\bar{g}_{\mu\nu} = \delta_{\mu\nu}$ with the flat $O(4)$-metric $\delta = \text{diag}(1,1,1,1)$. Such an expansion works well, if the full dynamical space-time is asymptotically flat, and no regimes with strong curvatures are present. Consequently, the results in the present work obtained for the spectral properties of gravity rely on this assumption.

The flat background choice comes with considerable technical advantages. In particular it allows for the definition of Fourier transforms and allows us to compute momentum-dependent vertices,

$$\Gamma^{(n)}(\boldsymbol{p}) = \left(\prod_{j=1}^{n}\int \mathrm{d}^4 x_j\, e^{\mathrm{i}x_j^{\mu}p_j^{\mu}}\right)\Gamma^{(n)}[\delta_{\mu\nu}, 0](\boldsymbol{x}). \tag{5}$$

Here, $\boldsymbol{p} = (p_1, \ldots, p_n)$. The vertices in (5) are computed within the fRG-approach to quantum gravity discussed below. Owing to the flat background the respective fRG-flow equations are standard momentum loops and hence can be solved within the well-developed computational machinery of fRG-computations in quantum field theories. Moreover, it facilitates the discussion of the Wick rotation to Minkowski space as we can resort to standard spectral properties. As already discussed before, this does not resolve the problem of a Wick rotation in the presence of a dynamical metric. However, the results here may also shed some light into this intricate challenge.

In the fRG approach to quantum gravity, the theory is regularised with an IR cutoff that suppresses quantum fluctuations with momenta $p^2 \lesssim k^2$. This cutoff is successively lowered and finally removed. The respective flow equation for the IR regularised effective action $\Gamma_k$ is given by [69–71]

$$\partial_t \Gamma_k[\bar{g}_{\mu\nu}, \phi] = \frac{1}{2}\operatorname{Tr} \mathcal{G}_k[\bar{g}_{\mu\nu}, \phi]\, \partial_t R_k[\bar{g}_{\mu\nu}], \tag{6}$$

with

$$\mathcal{G}_{\phi_{i_1}\phi_{i_2}, k}[\bar{g}_{\mu\nu}, \phi] = \left[\frac{1}{\Gamma_k^{(\phi\phi)}[\bar{g}_{\mu\nu}, \phi] + R_k[\bar{g}_{\mu\nu}]}\right]_{\phi_{i_1}\phi_{i_2}}. \tag{7}$$

Here, $R_k$ is a regulator that implements the suppression of IR modes and $t = \log k/k_{\text{ref}}$ is the (negative) RG-time with the reference scale $k_{\text{ref}}$ that is at our disposal. Note that the second derivative of the effective action with respect to the fluctuation field enters in (6). The flow equations for $\Gamma_k^{(n)}$ are obtained by $n$-derivatives w.r.t. the fluctuation field $\phi$. For more details on the fRG-approach to quantum gravity see [1–14].

We extract the momentum-dependent fluctuation graviton propagator from the flow of the two-point function, and the momentum-dependent flow of the fluctuation Newton coupling from the flow of the three-point function. The corresponding flow equations are diagrammatically depicted in Fig. 2. The regulator used for the numerical computations is a Litim-type regulator, see App. B.

$$\partial_t \mathbf{\Gamma}^{(2h)} = -\frac{1}{2} \quad \text{(diagram)} \quad + \quad \text{(diagram)} \quad - 2 \quad \text{(diagram)}$$

$$\partial_t \mathbf{\Gamma}^{(3h)} = -\frac{1}{2} \quad \text{(diagram)} \quad + 3 \quad \text{(diagram)} \quad - 3 \quad \text{(diagram)} \quad + 6 \quad \text{(diagram)}$$

Figure 2: Diagrammatic representation of the flows of the fluctuation graviton two- and three-point functions, from which we extract the flow of the fluctuation graviton propagator and the flow of the physical Newton coupling, respectively. The latter relates to the flow of the background graviton propagator. Double blue lines represent graviton propagators, red single lines ghost propagators, and the cross stands for a regulator insertion.

## 2.2 Projection on momentum-dependent couplings

We now describe the fRG setup for the solution of the momentum-dependent vertices as defined in (4) and (5) within a flat background. These numerical computations require a truncation of the effective action to a finite set of vertices. The present work builds on results and flows in [45], where the momentum dependence of the two-, there-, and four-point graviton vertices have been taken into account, as well as that of the graviton-ghost sector. Further works including momentum dependences can be found in [46–48, 55–58, 72–75], for a review see [14].

The general tensor structure of the vertices is furthermore reduced to the Einstein-Hilbert tensor structures,

$$\mathcal{T}_{\text{EH}}^{(\phi_{i_1} \cdots \phi_{i_n})}(\boldsymbol{p}; \Lambda_n) = S_{\text{EH}}^{(\phi_{i_1} \cdots \phi_{i_n})}(\boldsymbol{p}; \Lambda_n)\big|_{G_{\text{N}} \to 1}. \tag{8}$$

In (8), $S_{\text{EH}}^{(\phi_{i_1} \cdots \phi_{i_n})}$ is the $n$th derivative of the Einstein-Hilbert action (1). We send $G_{\text{N}} \to 1$ in order to remove the dependence on $G_{\text{N}}$. The $\Lambda_n$ are the (running) coefficients of the tensor structure arising from the cosmological-constant term. These coefficients can be understood as avatars of the cosmological constant. Guided by the results in [45], we use the further approximation $\Lambda_n \approx 0$, detailed below.

In the $n$-point vertices, the above tensor structures are multiplied by a scalar vertex dressing that depends on $\boldsymbol{p}$. Here we only consider the dressing with a dependence on the average momentum $\bar{p}$,

$$\bar{p}^2 = \frac{\boldsymbol{p}^2}{n}. \tag{9}$$

As it is multiplying the Einstein-Hilbert tensor structure, it can be understood as a power of an avatar $G_n(\bar{p})$ of the Newton coupling, multiplied with $\sqrt{Z_{\phi_i,k}(p_i)}$ for each leg. Here, the $Z_{\phi_i,k}(p_i)$ are the momentum-dependent wave-function renormalisations of the fields $\phi_i$. They relate to the anomalous dimensions of the fields $\phi_i$ via

$$\eta_{\phi_i}(p) = -\partial_t \ln Z_{\phi_i,k}(p). \tag{10}$$

The anomalous dimension $\eta_{\phi_i}$ are $k$-dependent just as the wave-function renormalisations $Z_{\phi_i,k}$ but we choose to suppress the index $k$ for convenience of notation. Note that $Z_h$ and $\eta_h$ are in general tensorial quantities and we choose a uniform wave-function renormalisation

for the graviton. In summary, this leads us an ansatz for the $n$-point functions of the effective action given by

$$\Gamma_k^{(\phi_{i_1}\dots\phi_{i_n})}(\boldsymbol{p}) = \left(\prod_{j=1}^{n} Z_{\phi_{i_j},k}^{\frac{1}{2}}(p_j)\right) G_n^{\frac{n}{2}-1}(\bar{p})(S_{\mathrm{EH}} + S_{\mathrm{gf}} + S_{\mathrm{gh}})^{(\phi_{i_1}\dots\phi_{i_n})}\Big|_{G_{\mathrm{N}}\to 1}. \tag{11}$$

The first term in the second line is precisely the tensor structure defined in (8). For $n > 2$, there are no contributions from the gauge-fixing term, and for $n > 3$, also ghost-graviton contributions are absent.

The $G_n(\bar{p})$ are running avatars of the Newton coupling for each $n$-point function. The flows of $\Gamma^{(n)}$ or that of the $G_n(\bar{p})$ are obtained within an evaluation of $\partial_t \Gamma^{(n)}$ at a symmetric point, $p_i^2 = \bar{p}^2$, where for the 3-point function we choose

$$p_i \cdot p_j = \frac{1}{2}(3\delta_{ij} - 1)\bar{p}^2. \tag{12}$$

We work with the dimensionless versions of $G_n$ and $\Lambda_n$, which are given by

$$g_n(p) = k^2 G_n(p), \qquad\qquad \lambda_n = \frac{\Lambda_n}{k^2}. \tag{13}$$

Here and in the following, we drop the bar on the momentum argument of $g_n$ but it is understood that we consider the average momentum flow through the vertex, see (9). More details on the projection procedure (contraction of the tensor structure) and the results can be found in [14, 45].

For the analysis of the spectral properties of the graviton presented here, we consider additional approximations that are guided by the results in [45]. There, the UV-fixed point as well as full UV-IR trajectories with momentum dependences for two-, three- and four-point functions have been considered. While not being identical, the avatars $g_n(\bar{p})$ of the Newton couplings showed similar $\bar{p}$- and $k$-dependences.

We assume a vanishing cosmological constant, which in the present approximation entails $\lambda_n = 0$ at vanishing cutoff scale, $k = 0$. We know from [45] that the flow is not dominantly driven by the $\lambda_n$ and for the sake of simplicity we use $\lambda_n(k) = 0$. Similarly, we use that the momentum-dependence of the ghost, while present, is only of quantitative interest. We use a vanishing ghost anomalous dimension, $\eta_c(p) \approx 0$. In summary, we compute the flows of

$$Z_{h,k}(p), \qquad\qquad g_{3,k}(p) = g_k(p), \tag{14a}$$

with vanishing $\lambda_n$ and $\eta_c$. Furthermore, we identify all avatars of the Newton coupling with $g_{3,k}$,

$$g_{n,k}(p) = g_k(p), \tag{14b}$$

which leaves us with a unique momentum- and cutoff-dependent Newton coupling $G_k(p)$, and a unique physical Newton coupling $G_{\mathrm{N}}(p)$ with

$$G_{\mathrm{N}}(p) = G_{k=0}(p), \quad \text{where} \quad G_k(p) = \frac{g_k(p)}{k^2}. \tag{14c}$$

We emphasise that *physical* simply refers to the physical limit $k \to 0$.

We consider RG-trajectories with classical IR scaling. Together with the definitions (14) and in particular (14c), this implies that the classical Newton coupling in (1) is nothing but the physical one at vanishing momentum, $G_{\mathrm{N}} = G_{\mathrm{N}}(0)$, which also defines the Planck mass,

$$M_{\mathrm{pl}}^2 = \frac{1}{G_{\mathrm{N}}(p=0)}. \tag{15}$$

Within the approximation described above, we can access the full momentum- and cutoff-dependent fluctuation propagator via $Z_{h,k}(p)$ as well as the three-graviton coupling $g_k(p)$. Here, $p = \bar{p}$ is the average momentum flow through the vertex, see (9). The respective flow is evaluated at the symmetric point (12). The flows are integrated from the initial condition close to the UV fixed point to the physical theory for $k \to 0$. The diagrams contributing to the respective two- and three-point function flows are displayed in Fig. 2. For the wave-function renormalisation we choose the initial condition $Z_{h,\Lambda} = \text{const.}$ and fix $Z_{h,k=0}(p = 0) = 1$. Here $\Lambda$ is the initial scale that we send to infinity and the constant initial conditions depends on $\Lambda$.

# 3 The graviton spectral function

The Euclidean fluctuation approach in Sec. 2 within the approximations discussed in Sec. 2.2 provides us with results for the momentum- and cutoff-dependent fluctuation field propagator and Newton coupling. For $k \to 0$ we approach the physical theory. This already allows us to discuss the properties of the physical correlation functions in momentum space.

In particular, it also gives access to the question, whether an identification of momentum-dependences at $k = 0$ and cutoff dependences at $p = 0$ is at least working qualitatively. Such an identification underlies many physics studies in asymptotic safety, most of which only provide cutoff dependences and not momentum dependences. While not being at the heart of the current work, the Euclidean momentum dependences provided here are hence very important for the physics interpretation of these cutoff scale studies. Even more importantly, the current results, as well as those already provided in [14, 45–48] can be used as input for the direct computation of scattering vertices for general momentum configurations, $S$-matrix elements, and asymptotically safe cosmology. These interesting applications are left to future work.

Here we aim at the reconstruction of the graviton spectral function from the numerical Euclidean data of the graviton propagator, for our results see Fig. 1. Such reconstructions based on numerical data with statistical and systematic errors are typically ill-conditioned problems. Moreover, for (unphysical) gauge fields they also require the additional key assumption that such a spectral representation exists. In gravity, this is further complicated by the intricacies of the Wick rotation. The considerations and numerical reconstructions here are based on these assumptions, a detailed investigation of the difficulties of the reconstruction for numerical data in the context of QCD can be found in [37]. Here, we follow the discussion there and extend it to positivity and normalisability of spectral functions in the presence of anomalous UV and IR momentum scalings. While most of the respective properties, in particular the UV ones, are well-known, the present work is to our knowledge the first comprehensive application to quantum gravity.

Time-ordered propagators $\mathcal{G}_F(x, y) = \langle T\phi(x)\phi(y)\rangle - \langle\phi(x)\rangle\langle\phi(y)\rangle$ of physical fields (asymptotic states) in Minkowski space have a Källén–Lehmann (KL) spectral representation. In momentum space, it is given by

$$\mathcal{G}_F(p_0) = i \int\limits_0^\infty \frac{d\lambda}{\pi} \frac{\lambda\,\rho(\lambda)}{p_0^2 - \lambda^2 + i\epsilon}\,, \tag{16}$$

with the spectral function $\rho(\lambda)$. In (16), the restriction to positive frequencies in the integral follows from the antisymmetry of the spectral function

$$\rho(\lambda) = -\rho(-\lambda)\,. \tag{17}$$

A simple example is provided by the classical spectral function $\rho_{cl}$ of a particle with pole mass

$m_{\text{pol}}$,

$$\rho_{\text{cl}}(\lambda) = \frac{\pi}{\lambda}\Big[\delta(\lambda - m_{\text{pol}}) - \delta(\lambda + m_{\text{pol}})\Big]. \tag{18}$$

Inserting (18) in (16) leads to the classical Feynman propagator $\mathcal{G}_{\text{F}}(p_0) = \text{i}/(p_0^2 - m_{\text{pol}}^2 + \text{i}\epsilon)$. The KL-representation in (16) constructs the full propagator in terms of a spectral integral over 'on-shell' propagators $1/(p_0^2 - \lambda^2 + \text{i}\epsilon)$ of states with pole masses $\lambda$. If properly normalised, the total spectral weight of all states is unity,

$$\int_0^\infty \frac{\text{d}\lambda}{\pi}\, \lambda\, \rho(\lambda) = 1\,. \tag{19}$$

The spectral sum rule in (19) holds for the spectral function of asymptotic states and also encodes the unitarity of the theory. In general, the propagator of the fundamental fields in the theory is subject to renormalisation and its amplitude can be changed by an RG equation. In this case, one first has to define renormalisation group invariant fields to apply the above arguments. For gauge fields, the discussion is even more intricate as detailed below.

The condition (19) entails that the decay of the spectral function for asymptotically large spectral values has to be faster than $1/\lambda^2$. The latter decay is the canonical one, as the momentum-dimension of the spectral function is that of the propagator: $-2$. The classical spectral function is a $\delta$-function, and vanishes identically for $\lambda > m_{\text{pol}}$. In turn, scattering events for $\lambda > m_{\text{pol}}$ induce a spectral tail, which indeed decays faster than $1/\lambda^2$. However, if the propagator shows an anomalous momentum scaling for large momenta, this analysis is more intricate and is detailed below. This case with anomalous scaling applies to gauge theories, and in particular to the graviton.

The reconstruction of the spectral function is done with the Euclidean propagator $\mathcal{G}(p_0)$ for Euclidean momenta $p_0$. In the Euclidean branch this spectral representation of $\mathcal{G}(p_0) = \text{i}\,\mathcal{G}_{\text{F}}(\text{i}\,p_0)$ is given by

$$\mathcal{G}(p_0) = \int_0^\infty \frac{\text{d}\lambda}{\pi} \frac{\lambda\, \rho(\lambda)}{\lambda^2 + p_0^2}\,. \tag{20}$$

Equivalently, the spectral function can be obtained from the Euclidean propagator by means of an analytic continuation,

$$\rho(\omega) = 2\,\mathfrak{I}\,\mathcal{G}\big(-\text{i}(\omega + \text{i}0^+)\big)\,, \tag{21}$$

i.e. from the discontinuity of the propagator. Inserting the limit on the right-hand side of (21) in (20) leads to a $\delta$-function from the KL kernel, and the spectral integral can readily be performed, leading to the left-hand side of (21). This concludes the brief introduction to the KL representation.

## 3.1 Properties of the graviton spectral function

Gauge fields such as the graviton and the gluon are not directly linked to asymptotic states. Therefore, they do not necessarily enjoy a spectral representation. While the photon in QED is believed to have a spectral representation, this is currently a debated subject in QCD, see e.g. [37, 76] and references therein. Possible extensions of (16) include complex conjugated poles, which are not considered here, as they may signal the loss of unitarity. However, for recent discussions of such an extension including the question of unitarity see e.g. [77–80].

In QCD, it is the confining nature at large distances that complicates the matter, and in particular the relation to asymptotic states. In asymptotically safe quantum gravity, it is the strongly-correlated UV fixed-point regime that complicates spectral considerations, even if leaving aside the intricacies of spectral representation in the presence of dynamical metrics. In turn, in the IR, gravity is well-described by the classical Einstein-Hilbert action (1) and we expect spectral properties in the IR similar to that of the photon: a massless $\delta$-function with a scattering tail. This expectation is well-supported by the observations at LIGO [81]. In summary, this suggests spectral properties of the graviton that are 'photon-like' in the IR and 'gluon-like' for the Planck-scale and beyond.

As discussed in Sec. 2, we consider AS gravity within an expansion about a flat background. In this setup, gravity is classical for large distances since the UV-IR trajectories approach classical scaling in the IR [45]. In this limit, similarly to QED, gravity is weakly coupled and may enjoy a spectral representation.

The full propagator can be decomposed in its different components, leaving us with a traceless-transverse tensor as well as vector and scalar components. In the present approximation based on the Einstein-Hilbert tensor structure, all components are related and it suffices to discuss the spectral representation of one of them. Here, we concentrate on the spectral function of the traceless-transverse part $\mathcal{G}_{hh,\mathrm{TT}}$ of the graviton propagators in a flat background, also considered in [45]. We parametrise $\mathcal{G}_{hh,\mathrm{TT}}$ with the TT-projection operator $\Pi_{\mathrm{TT}}(p)$ in App. C

$$\mathcal{G}_{hh,\mathrm{TT}}(p) = \mathcal{G}_{hh}(p)\Pi_{\mathrm{TT}}(p),$$

$$\mathcal{G}_{\bar{g}\bar{g},\mathrm{TT}}(p) = \mathcal{G}_{\bar{g}\bar{g}}(p)\Pi_{\mathrm{TT}}(p), \tag{22}$$

with the scalar parts $\mathcal{G}_{hh}(p)$ and $\mathcal{G}_{\bar{g}\bar{g}}(p)$ of the fluctuation and background graviton respectively. Both scalar propagators are assumed to have a KL representation (20) with spectral functions $\rho_h(\lambda)$ and $\rho_{\bar{g}}(\lambda)$.

Importantly, both the analytic IR and the UV tail of the Euclidean propagators can be used to analytically determine the spectral functions $\rho(\lambda)$ for the asymptotic regimes $\lambda \to 0$ and $\lambda \to \infty$, see [37]. In the UV this is related to the well-known Oehme-Zimmermann super-convergence relation [82,83]. We recall the argument here, adapted to the AS graviton. We consider dimensionless propagators and momenta, which are rescaled by appropriate powers of the Planck mass, see (15). This leads us to dimensionless momenta and spectral parameters,

$$\hat{p}^2 = \frac{p^2}{M_{\mathrm{pl}}^2}, \qquad\qquad \hat{\lambda} = \frac{\lambda}{M_{\mathrm{pl}}}, \tag{23a}$$

and dimensionless propagators and spectral functions,

$$\hat{\mathcal{G}}(\hat{p}) = M_{\mathrm{pl}}^2 \mathcal{G}(p), \qquad\qquad \hat{\rho}(\hat{\lambda}) = M_{\mathrm{pl}}^2 \rho(\lambda). \tag{23b}$$

With (23) the UV limit of the dimensionless propagators reads

$$\lim_{\hat{p}^2 \to \infty} \hat{\mathcal{G}}(\hat{p}) = \frac{Z^{\mathrm{UV}}}{\hat{p}^{2\left(1-\frac{\eta}{2}\right)}} \frac{1}{(\log \hat{p}^2)^{\bar{\gamma}}}, \tag{24}$$

where the $Z^{\mathrm{UV}}$'s are dimensionless normalisations and the $\eta$'s are the anomalous dimensions of the fluctuation graviton $h_{\mu\nu}$ and the background graviton $\bar{g}_{\mu\nu}$, and $\bar{\gamma}$ is non-vanishing for marginal scalings. Eq. (24) is a general asymptotic form that includes a monomial behaviour as well as a logarithmic cut. The decay behaviour (24) present in asymptotically safe gravity is different to some non-local gravity theories that feature an exponential decay behaviour.

The anomalous dimensions in (24) are precisely given by the anomalous dimension at $k \to \infty$ and $p = 0$ if the theory is momentum local [48]. If the theory is not momentum local and the relation between $\eta_{k \to \infty}(p = 0)$ and the fall-off of the propagator at $k = 0$ is more intricate. In the current approximation, the theory is momentum local and the anomalous dimensions take the values

$$\eta_h \approx 1.03, \qquad\qquad Z_h^{\text{UV}} = 0.64,$$

$$\eta_{\bar{g}} = -2, \qquad\qquad Z_{\bar{g}}^{\text{UV}} = 24.1, \tag{25}$$

with $\bar{\gamma} = 0$. While the anomalous dimension of the fluctuation graviton $\eta_h$ is a dynamical quantity and depends on the approximation, the background anomalous dimension is uniquely fixed by asymptotic safety: It is linked to the $\beta$-function $\beta_{\text{N}}$ of the background Newton coupling with $\eta_{\bar{g}} = \beta_{\text{N}} - 2$. At the fixed point $\beta_{\text{N}}$ is vanishing by definition and hence $\eta_{\bar{g}} = -2$ follows. The value of $\eta_h$ is approximation-dependent but one finds $\eta_h > 0$.

Eq.(25) already shows an interesting difference between the graviton fields: The background graviton has a negative anomalous dimension while the fluctuation graviton has a sizeable positive anomalous dimension.

The computation of $\eta_h$ and the underlying approximation is explained later, and is done in the de-Donder type gauge (69) with $\alpha = 0$ and $\beta = 1$, given in App. A. The large momentum value in (25) has been computed in [45] within a rather elaborate approximation. The approximation here is a variant of that put forward in [45], and utilises the flows derived there.

For the general discussion as well as the consideration of subleading UV- and IR-momentum dependences in the graviton propagators we also take into account a potential logarithmic running. This is known from resummed perturbation theory, where $\bar{\gamma}$ is given by the ratio of the anomalous dimension and the $\beta$-function of the running coupling,

$$\bar{\gamma} = \frac{\eta}{\beta}. \tag{26}$$

In summary, (24) allows us to discuss the UV-asymptotics of the spectral function of a given propagator. Moreover, it also can be used for the IR asymptotics, $p \to 0$, where it gives access to the IR asymptotics of the spectral function.

### 3.1.1 Spectral function of the background graviton

We first discuss the UV limit of the background graviton. The argument follows closely that for the Oehme-Zimmermann super-convergence relation in QCD. We show that the spectral function of the background graviton is negative for large spectral values and its total spectral weight vanishes,

$$\int_0^\infty d\lambda \, \lambda \, \rho_{\bar{g}}(\lambda) = 0. \tag{27}$$

Hence, in contradistinction to the spectral sum rule (19) related to unitarity of the theory, the spectral sum rule (27) enforces a vanishing spectral sum. Note also that (27) necessitates a spectral function $\rho_{\bar{g}}(\lambda)$ that is both positive and negative for some $\lambda$.

For proving (27), we consider asymptotically large Euclidean momenta as compared to the Planck mass. It is convenient to study the asymptotic properties in terms of the dimensionless quantities defined in (23) within the limit $\hat{p} = p/M_{\text{pl}} \to \infty$. In this limit the propagator of the background graviton decays with,

$$\lim_{\hat{p} \to \infty} \hat{\mathcal{G}}_{\bar{g}\bar{g}}(\hat{p}) = \frac{Z_{\bar{g}}^{\text{UV}}}{\hat{p}^{2\left(1 - \frac{\eta_{\bar{g}}}{2}\right)}}, \tag{28}$$

with $\eta_{\bar{g}} \to -2$, see (24) and (25). The UV asymptotics in (28) allows us to determine the spectral representation for $\hat{\lambda} \to \infty$. With the definition (21) we get

$$\lim_{\hat{\lambda} \to \infty} \hat{\rho}_{\bar{g}}(\hat{\lambda}) = 2 \sin\left[\frac{\pi}{2}\eta_{\bar{g}}\right] \frac{Z_{\bar{g}}^{\mathrm{UV}}}{\hat{\lambda}^{2-\eta_{\bar{g}}}}. \tag{29}$$

Evidently, for $\eta_{\bar{g}} < 0$, the spectral function decays more rapidly as $1/\hat{\lambda}^2$. Moreover, if we approach the UV-fixed point scaling with $\eta_{\bar{g}} \to -2$, the right-hand side in (29) vanishes. Then, the spectral function $\rho_{\bar{g}}$ decays either more rapidly than $1/\hat{\lambda}^4$ or does vanish identically. In summary, (29) guarantees that all spectral integrals in the following are finite.

Now we split the spectral integral in (20) for the background graviton propagator into an asymptotic UV part with spectral values $\hat{\lambda} \geq \sqrt{\hat{p}}$, and the respective IR part,

$$\mathcal{G}_{\bar{g}\bar{g}}(\hat{p}) = \int_0^{\sqrt{\hat{p}}} \frac{\mathrm{d}\hat{\lambda}}{\pi} \frac{\hat{\lambda}\hat{\rho}_{\bar{g}}(\hat{\lambda})}{\hat{\lambda}^2 + \hat{p}^2} + \int_{\sqrt{\hat{p}}}^{\infty} \frac{\mathrm{d}\hat{\lambda}}{\pi} \frac{\hat{\lambda}\hat{\rho}_{\bar{g}}(\hat{\lambda})}{\hat{\lambda}^2 + \hat{p}^2}. \tag{30}$$

Let us first discuss the second term on the right-hand side of (30): For $\sqrt{\hat{p}} \to \infty$ only the asymptotic limit of the spectral function in (29) enters, and the term decays faster than $1/\hat{p}^2$. For $\eta_{\bar{g}} \in (-2, 0)$, we find

$$\lim_{\hat{p} \to \infty} \left| \int_{\sqrt{\hat{p}}}^{\infty} \frac{\mathrm{d}\hat{\lambda}}{\pi} \frac{\hat{\lambda}\hat{\rho}_{\bar{g}}(\hat{\lambda})}{\hat{\lambda}^2 + \hat{p}^2} \right| \leq \frac{C}{\hat{p}^{2-\frac{\eta_{\bar{g}}}{2}}}. \tag{31}$$

For $\eta_{\bar{g}} = 0$, the respective fall-off behaviour is $C \log(\hat{p})\hat{p}^{-2}$, and, for $\eta_{\bar{g}} \in (0, 2)$, it is $C\hat{p}^{-2+\eta_{\bar{g}}}$. Accordingly, for our case of interest with $\eta_{\bar{g}} \in (-2, 0)$, also the first term on the right-hand side in (30) has to decay at least with $1/\hat{p}^{(2-\eta_{\bar{g}}/2)}$, in order to guarantee the limit (28) in combination with (31). For example, for the fixed point scaling with $\eta_{\bar{g}} = -2$ this amounts to a decay with $1/\hat{p}^3$.

The first term in (30) can be rewritten as

$$\int_0^{\sqrt{\hat{p}}} \mathrm{d}\hat{\lambda} \frac{\hat{\lambda}\hat{\rho}_{\bar{g}}(\hat{\lambda})}{\hat{\lambda}^2 + \hat{p}^2} = \frac{1}{\hat{p}^2} \int_0^{\sqrt{\hat{p}}} \mathrm{d}\hat{\lambda}\hat{\lambda} \frac{\hat{\rho}_{\bar{g}}(\hat{\lambda})}{1 + \frac{\hat{\lambda}^2}{\hat{p}^2}}, \tag{32}$$

where we have dropped the $1/\pi$-term, as the total normalisation is not relevant for the present discussion. Now we use $\hat{\lambda}^2 \leq \hat{p}$ in (32) due to the upper bound of the integration. Hence, in the limit $\hat{p} \to \infty$, the $\hat{\lambda}^2$-part in the denominator in (32) can be dropped to leading order. Accordingly, in this limit, we are led to

$$\frac{1}{\hat{p}^2} \int_0^{\sqrt{\hat{p}}} \mathrm{d}\hat{\lambda}\hat{\lambda} \frac{\hat{\rho}_{\bar{g}}(\hat{\lambda})}{1 + \frac{\hat{\lambda}^2}{\hat{p}^2}} \xrightarrow{\hat{p} \to \infty} \frac{1}{\hat{p}^2} \int_0^{\sqrt{\hat{p}}} \mathrm{d}\hat{\lambda}\hat{\lambda}\hat{\rho}_{\bar{g}}(\hat{\lambda}). \tag{33}$$

The prefactor only decays with $1/\hat{p}^2$ for $\hat{p} \to \infty$. This entails that for $\eta_{\bar{g}} < 0$, the spectral integral in (33) has to decay at least as $\hat{p}^{\eta_{\bar{g}}/2}$ in order to be compatible with (28) for the background propagator. This leads us to

$$\lim_{\hat{p} \to \infty} \int_0^{\sqrt{\hat{p}}} \mathrm{d}\hat{\lambda}\hat{\lambda}\hat{\rho}_{\bar{g}}(\lambda) = 0, \qquad \text{for} \qquad \eta_{\bar{g}} < 0. \tag{34}$$

Eq. (34) is nothing but the Oehme-Zimmermann super-convergence sum rule (27).

This property holds for any field with a UV scaling with $\eta < 0$. In particular, we conclude that, if the background graviton admits a spectral representation, its spectral function, $\rho_{\bar{g}}$, has a vanishing total spectral weight, see (27). This also implies negative parts for $\rho_{\bar{g}}$, which is confirmed in the explicit computation, see Fig. 1.

We emphasise again that the property (27) does not entail unitary violations. The same property holds true for the background gluon $\bar{A}_\mu$ in QCD, where the full gauge field $A_\mu$ is split into a background field $\bar{A}_\mu$ and a fluctuation field $a_\mu$ with the linear split $A_\mu = \bar{A}_\mu + a_\mu$. If it has a spectral representation, it has the property (27) due to its anomalous dimension being negative,

$$\eta_{\bar{A}} = \beta_{g_s^2} = -\frac{g_s^2}{16\pi^2}\frac{22}{3} < 0, \tag{35}$$

with the running strong coupling $g_s(p/\Lambda_{\text{QCD}})$. For large momenta the coupling tends to zero due to asymptotic freedom and hence $\eta_{\bar{A}} \to 0$. The ratio $\bar{\gamma}$ of anomalous dimension $\eta_{\bar{A}}$ and $\beta$-function $\beta_{g_s^2}$ is unity, $\bar{\gamma}_{\bar{A}} = 1$ and we are left with a logarithmic running $1/p^2 1/(\log p/\Lambda_{\text{QCD}})$ of the propagator. In this case, the decay in (31) solely arises from the respective logarithms for $\bar{\gamma} > 0$.

Similarly to gravity, the anomalous dimension of the graviton is identical to the (anomalous) part of the $\beta$-function of the coupling. Note that in QCD this property is even more peculiar: the spectral function is negative in a regime, where the theory is asymptotically free. In any case, this analogy makes clear that a negative spectral function for an unphysical gauge boson does not entail a lack of unitarity for the theory. While unitary of QCD has not been proven rigorously, it is commonly assumed that it is present. However, let us also add that negative spectral functions do not facilitate unitarity proofs or arguments either.

### 3.1.2 Spectral functions for large spectral values & normalisation

Now we use the UV-leading term of the propagators for both, the fluctuation graviton and the background graviton in (24) for determining the asymptotic form for both spectral functions. Here we expect a qualitative difference between gravity and QCD/Yang-Mills theory. In the latter theory, the fluctuation gluon has the same vanishing spectral weight property of (27) (in the Landau-DeWitt gauge) due to the negative anomalous dimension $\eta_a$ of the fluctuation gluon $a_\mu$,

$$\eta_a = -\frac{g_s^2}{16\pi^2}(13 - 3\xi) < 0, \tag{36}$$

for $\xi < 13/3$, where $\xi$ is the gauge-fixing parameter. As for the background propagator we have $\eta_a \to 0$ for $p/\Lambda_{\text{QCD}} \to \infty$. The resummed logarithmic running has the power

$$\bar{\gamma}_a = \frac{13}{22}, \tag{37}$$

for the Landau gauge with $\xi = 0$. For more details and the discussions of general covariant gauges we refer the reader to e.g. [84] and references therein. As for the background gluon, the fluctuation gluon propagator decays more rapidly as $1/p^2$ and we arrive at (27): both gluon propagators obey the sum rule (27), their total spectral weight vanishes.

In turn, in AS gravity the anomalous dimension of the fluctuation graviton is positive, $\eta_h > 0$, see (25). The respective computation in [45] as well as the present ones are done in the de-Donder type gauge (69) with $\alpha = 0$ and $\beta = 1$. The choice $\alpha = 0$ enforces the gauge strictly similar to the Landau-DeWitt gauge in QCD with gauge fixing parameter $\xi = 0$.

We now proceed with the analytic computation of the UV asymptotics of the spectral functions. Using (21) one obtains the asymptotic behaviour of the smooth part of the dimensionless spectral functions $\hat{\rho}$ in (23) with,

$$\lim_{\hat{\omega}\to\infty} \hat{\rho}(\hat{\omega}) = 2\frac{Z^{\text{UV}}}{\hat{\omega}^{2\left(1-\frac{\eta}{2}\right)}} \frac{1}{(\log\hat{\omega}^2)^{\bar{\gamma}}} \left(\sin\left[\frac{\pi}{2}\eta\right] - \cos\left[\frac{\pi}{2}\eta\right]\frac{\pi\bar{\gamma}}{\log\hat{\omega}^2}\right), \tag{38}$$

valid for the $\eta \in (\eta_h, \eta_{\bar{g}})$ considered here,

$$\eta \in (-2, 2). \tag{39}$$

In (38) we have dropped subleading terms in the logarithms with:

$$\sqrt{\pi^2/4 + (\log\hat{\omega}^2)^2} \to \log\hat{\omega}^2.$$

The lower limit in (39) comes from a constraint in the fixed-point theory, which admits scaling for all momenta. For $\eta < -2$ the fixed-point propagator is not plane-wave normalisable any more, it has no Fourier representation as it develops a non-integrable singularity at $p = 0$. The boundary value $\eta = -2$ is special and has to be evaluated with care. The upper limit is a more technical one, the approximation and regulators used in the present (and most other works) fails for $\eta > 2$, see [55]. Again the boundary value requires special attention.

Importantly, we can already conclude from our analysis that fields with $\eta \neq 0$ cannot describe asymptotic states: for $\eta < 0$ the spectral function necessarily has negative parts, while for $\eta > 0$ the spectral function is not (UV) normalisable, as the spectral function decays with less than $1/\hat{\lambda}^2$.

For $\eta = 0$ we are left with the dependence on $\bar{\gamma}$. In this case, as for $\eta = \pm 2$, the first term in (38) vanishes. For $\bar{\gamma} \neq 0$ we are left with the second term, triggered by the logarithmic running of the UV-asymptotics. This part, with $\eta = 0$ and $\bar{\gamma} > 0$, covers the QCD-behaviour. The UV asymptotics is given by

$$\lim_{\hat{\omega}\to\infty} \hat{\rho}(\hat{\omega}) = -\frac{2Z^{\text{UV}}}{\hat{\omega}^2} \frac{\pi\bar{\gamma}}{(\log\hat{\omega}^2)^{(1+\bar{\gamma})}}. \tag{40}$$

With (40), the total spectral weight is finite for $\bar{\gamma} > 0$. However, in this case, the spectral function necessarily has negative parts. In turn, for $\bar{\gamma} \leq 0$, the spectral weight is UV-divergent, and the spectral function cannot be normalised.

In summary, we have found that a spectral function has a vanishing total spectral weight for $\eta < 0$, see (27). In this case $\bar{\gamma}$ can be general. This property also holds true for $\eta = 0$ and $\bar{\gamma} > 0$,

$$\{\eta < 0 \text{ or } (\eta = 0 \wedge \bar{\gamma} > 0)\}: \quad \int_0^\infty d\lambda\, \lambda\, \rho(\lambda) = 0. \tag{41}$$

Then, the spectral function also has negative parts, and in particular, its UV asymptotics is negative in the range (39), see (38). This case applies to the spectral function of the background graviton, $\rho_{\bar{g}}$. We emphasise that this property is not at odds with unitarity for two reasons. First of all, it is a well-known property of the gluon in QCD (assuming the existence of a spectral representation). Secondly, the background graviton is not the graviton propagating in loop diagrams that contribute to the (unitary) S-matrix.

The graviton that is relevant for the latter processes in the S-matrix, is the fluctuation graviton. For the fluctuation graviton, the case $\eta > 0$ applies for the UV asymptotics. This

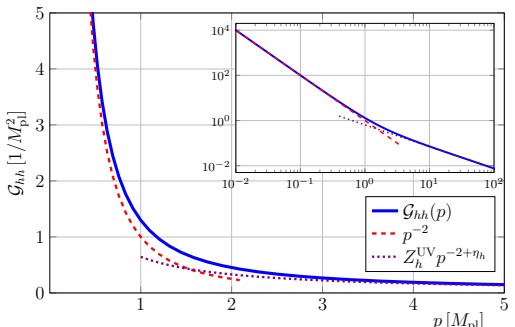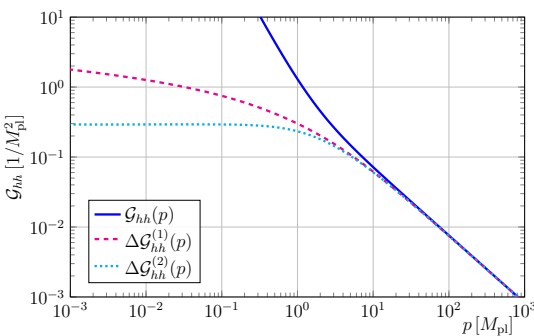

Figure 3: Momentum dependence of the Euclidean fluctuation graviton propagator $\mathcal{G}_{hh}(p)$. (left) Momentum dependence of the Euclidean fluctuation graviton propagator (blue, solid) with its IR and UV asymptotics. The UV asymptotic (violet, dotted) is given by $Z_h^{\mathrm{UV}} p^{-2+\eta_h}$ with $\eta_h = 1.03$ and $Z_h^{\mathrm{UV}} = 0.64$, see (25). The IR asymptotic (red, dashed) is simply the classical dispersion $1/p^2$. (right) Subleading contributions at small momenta in comparison to the full propagator (blue, solid). $\Delta\mathcal{G}_{hh}^{(1)}$ (magenta, dashed) carries a subleading log-like contribution, while $\Delta\mathcal{G}_{hh}^{(2)}$ (cyan, dotted) carries a constant contribution for small momenta.

entails that the UV tail of the spectral function is positive in the range (39). However, as discussed above, for $\eta_h > 0$ the spectral function cannot be normalised.

This UV analysis above does not imply that the spectral function $\rho_h$ is positive for all spectral values. However, we shall see later that this is indeed the case within the reconstruction, see also Fig. 1. This leads us with a positive, though not normalisable, spectral function $\rho_h > 0$. With the latter property of the fluctuation graviton one of the necessary condition for applying Cutkosky cutting rules, [85], is satisfied. This brings us closer to a reliable discussion of unitarity in asymptotic safety.

## 4 Euclidean correlation functions

With the setup discussed in Sec. 2, we now compute the Euclidean fluctuation graviton propagator, see Sec. 4.1, as well as the Euclidean coupling of the fluctuation three-point function for all momenta and cutoff scales, see Sec. 4.2. This is based on momentum-dependent results for the anomalous dimensions $\eta_h(p)$ and the $\beta$-function $\beta_{g_3}(p)$ in [45]. Here we provide, for the first time, the full physical momentum-dependence of the graviton two- and three-point function at vanishing cutoff scale. This also allows us to explicitly check the reliability of the identification of cutoff and momentum scales.

For the sake of simplicity, we use an analytical flow equation for the zero-momentum cutoff-dependent Newton coupling $g_k = g_k(p=0)$. This analytic flow equation is based on [45] with the approximations from (14) and $\eta_h = 0$. It takes the simple form,

$$\partial_t g_k = 2g_k\left(1 - \frac{g_k}{g^*}\right) = 2g_k - \frac{833 g_k^2}{285\pi}, \tag{42}$$

where $g^* = 570\pi/833$ is the UV fixed-point value. This flow equation has the solution

$$g_k = \frac{g^* k^2}{g^* M_{\mathrm{pl}}^2 + k^2}. \tag{43}$$

This solution is consistent with the fixed-point value in the UV, $g_{k\to\infty} = g^*$, and has the physical IR behaviour, $g_{k\to 0} = k^2/M_{\rm pl}^2$. This allows us to express the RG scale in units of the Planck mass.

On the trajectory (43), we evaluate the momentum-dependent graviton anomalous dimension $\eta_h(p)$ as well as the momentum-dependent three-point Newton coupling $g_k(p)$. We emphasise that both quantities are evaluated consistently on the trajectory (43) but we neglect any feedback on the trajectory itself. The impact of this approximation is subleading since the full trajectory, which can be obtained in an iterative procedure, exhibits the same qualitative features as (43).

## 4.1 Fluctuation graviton propagator

In this section, we present the Euclidean results for the propagator of the fluctuation graviton. We first discuss the details of the computation and the numerical results while we discuss analytic fits for the IR asymptotics in Sec. 4.1.1. The latter are important for the reconstruction of the spectral function.

The momentum-dependence of the Euclidean propagator is incorporated in the momentum-dependence of the anomalous dimension already computed in [45]. For the Euclidean scalar part $\mathcal{G}_{hh,k}(p)$ of the transverse-traceless mode (22) we parametrise the cutoff-dependent graviton propagator with

$$\mathcal{G}_{hh,k}(p) = \frac{1}{Z_{h,k}(p)\,p^2}\,. \tag{44}$$

The wave-function renormalisation is readily computed from the anomalous dimension $\eta_h(p^2)$, defined in (10). We emphasise that the anomalous dimension naturally also depends on the graviton couplings via the diagrams, see Fig. 2. With the definition (10), we obtain the physical wave-function renormalisation $Z_h(p)$, in the double limit $k \to 0$ and $\Lambda \to \infty$,

$$Z_h(p) = \lim_{\substack{k\to 0 \\ \Lambda\to\infty}} Z_{h,\Lambda}(p)\exp\Bigl(\int_k^\Lambda \frac{{\rm d}k'}{k'}\eta_{h,k'}(p)\Bigr)\,, \tag{45}$$

where we set $Z_{h,\Lambda} = {\rm const.}$ at a large cutoff scale and normalise $Z_h(p=0) = 1$. The computation of the fluctuation graviton anomalous dimension is detailed in App. E.

The result for the physical full momentum-dependent Euclidean graviton propagator $\mathcal{G}_{hh}(p)$ is presented in Sec. 3.1.2. The leading asymptotics of $\mathcal{G}_{hh}(p)$ are proportional to $1/p^2$ for small momenta, and $p^{\eta_h-2}$ for large momenta, where $\eta_h \approx 1.03$ is the graviton anomalous dimension at the UV fixed point and $p^2 = 0$, see (25).

### 4.1.1 IR asymptotics

The low-momentum asymptotic of $1/p^2$ captures the classical IR-regime: the theory approaches classical gravity with the Einstein-Hilbert action in (1). This does not exclude the presence of subleading features which may carry important physics. We access the subleading IR behaviour by subtracting the $1/p^2$-pole and introduce the difference propagator $\Delta\hat{\mathcal{G}}_{hh}^{(1)}$,

$$\Delta\hat{\mathcal{G}}_{hh}^{(1)}(p) = \hat{\mathcal{G}}_{hh}(\hat{p}) - \frac{1}{\hat{p}^2}\,. \tag{46}$$

This difference propagator is displayed with a red-dashed line in Sec. 3.1.2. As expected, the Euclidean propagator does indeed show non-trivial subleading behaviour introduced by

scatterings. It exhibits a log-like contribution, and for small momenta we find,

$$\lim_{\hat{p}\to 0} \Delta\hat{\mathcal{G}}^{(1)}_{hh}(\hat{p}) = -A_h \ln\hat{p}^2 + C_h\,, \tag{47}$$

with $A_h = \frac{7}{20\pi} \approx 0.11$ and $C_h \approx 0.29$. The coefficient $A_h$ is the prefactor of the $\hat{p}^4 \ln\hat{p}^2$ term in the two-point function. Accordingly, it can be computed by a $p^2$ derivative of the one-loop anomalous dimension at vanishing momentum, $A_h = -\partial_g \partial_{p^2} \eta_h(p)/2|_{g,p\to 0}$. The quantity thus relates to a $p^4$ derivative of the one-loop flow and is independent of the regulator, though still gauge dependent, see (85) in App. F for the full gauge-dependent result. The result in the gauge we use here agrees with effective field theory computations [86, 87].

For the reliability of the reconstruction it is beneficial to remove both the IR and UV asymptotics in terms of analytic functions. Then, the reconstruction only deals with the intermediate momentum (and spectral) values, which stabilises the construction. The analytic fits in the IR (UV) should not interfere with the UV (IR) behaviour, and should not introduce further structures. Note that these features are best (and most easily) implemented on the level of asymptotic spectral contributions. Here, we are also interested in Euclidean fits and stay in the Euclidean domain for the derivation of the analytic fits.

The subtraction with $1/\hat{p}^2$ in (46) satisfies these properties, as it is subleading in the UV due to $\eta_h > 0$. In turn, we cannot use the logarithmic and constant terms in (47) as an analytic IR fit for the subleading IR behaviour. Instead, we use the confluent hypergeometric function $U_{a,b}(\hat{p}^2)$, whose leading large-momentum asymptotic is $1/\hat{p}^{2a}$. For $b = 1$ and small momenta, it approaches

$$\lim_{\hat{p}\to 0} U_{a,1}(\hat{p}^2) = -\frac{1}{\Gamma(a)}\left(2\gamma + \frac{\Gamma'(a)}{\Gamma(a)} + \ln(\hat{p}^2)\right)\,, \tag{48}$$

where $\gamma$ is the Euler–Mascheroni constant and $\Gamma(z)$ the gamma function. Hence it shows the subleading IR-asymptotics in (47) with a cut at $\Re(p) = 0$. In particular, $U_{a,b}(\hat{p}^2)$ does not introduce any poles in the positive real half-plane. Moreover, for $a > 1 - \eta_h/2 \approx 0.485$ it is subleading in the UV. In summary, the hypergeometric functions $U_{a,1}$ also fulfil the requirement of not interfering with the UV behaviour of $\rho_h$, while simultaneously not introducing any additional structures, and are thus well suited to describe the log-like contribution at small momenta. For simplicity, we choose $a = 1$ and arrive at,

$$\Delta\hat{\mathcal{G}}^{(2)}_{hh}(\hat{p}) = \Delta\hat{\mathcal{G}}^{(1)}_{hh}(\hat{p}) - A_h U_{1,1}(\hat{p}^2)\,, \tag{49a}$$

where

$$U_{1,1}(\hat{p}^2) = e^{\hat{p}^2} \Gamma(0, \hat{p}^2)\,, \tag{49b}$$

with the upper incomplete gamma function $\Gamma(a,z) = \int_z^\infty dt\, t^{a-1} e^{-t}$. The subleading IR-asymptotics in (49) is depicted as the dotted cyan line in Sec. 3.1.2. In summary, the two IR subtractions leave us with a constant contribution remaining for small momenta.

The spectral function of these asymptotic IR fits is readily computed, which leaves us only with a reconstruction task of the remaining part of the propagator, $\Delta\hat{\mathcal{G}}^{(2)}_{hh}$. This is done by a fit of Breit-Wigner (BW) structures as well as an analytic UV-asymptotic $\rho_h^{\text{UV}}$. This is detailed in Sec. 5, the resulting spectral function is discussed in Sec. 5.2.

## 4.2 Newton coupling

In this section, we present the Euclidean results for the physical momentum-dependent Newton coupling $G_{\text{N}}(p) = G_{k=0}(p)$, which is derived from the transverse-traceless part of the fluctuation three-graviton vertex.

In our approximation, we only retain the dependence of the Newton coupling on the average momentum flowing through the vertex, see (9), and we evaluate the flow of the graviton three-point function at the momentum symmetric point, see (12). We feed the dependence of the average momentum back on the right-hand side of the flow, see Fig. 2. Note that different combinations of external and loop momenta run through the vertices in the diagrams but the coupling only depends on the average momentum. Furthermore, we use that the loop momentum $q$ is bounded by the cutoff scale, $q^2 \lesssim k^2$, and that the diagrams give subleading contributions for $p_i^2 \gg k^2$. This implies that through all vertices, we have an average momentum flow of the order of $\bar{p}^2 = 1/3(p_1^2 + p_3^2 + p_2^2)$ and we approximate $g_k(p_i, q) \approx g_k(\bar{p})$. More details can be found in [14, 45]. In summary, this leads us to

$$\partial_t g_k(\hat{p}) - \hat{p}\, g'_k(\hat{p}) = \left(2 + \eta_g(\hat{p})\right) g_k(\hat{p}), \tag{50}$$

with the dimensionless momentum $\hat{p} = p/k$ and the anomalous dimension $\eta_g$ of the flow of the graviton three-point function, see App. D for details. Eq. (50) is integrated for given data of $\eta_g(p)$. The resulting momentum-dependent Newton coupling at vanishing cutoff scale is given by the blue solid line in Fig. 4. Together with the explicit depiction of the momentum dependence of the fluctuation propagator, it is a key Euclidean result of the present work. It encodes, for the first time, the full momentum-dependence of the scattering coupling of three gravitons in the physical cutoff limit $k \to 0$ with

$$G_N(p) = G_{k=0}(p). \tag{51}$$

The coupling $G_N(p)$ shows a flat classical IR regime, and exhibits a slight increase in strength between about 1 and 2 Planck masses, before decaying with $1/p^2$. Whether or not this increase about the Planck scale is a physics feature or a truncation artefact remains to be seen within improved approximations.

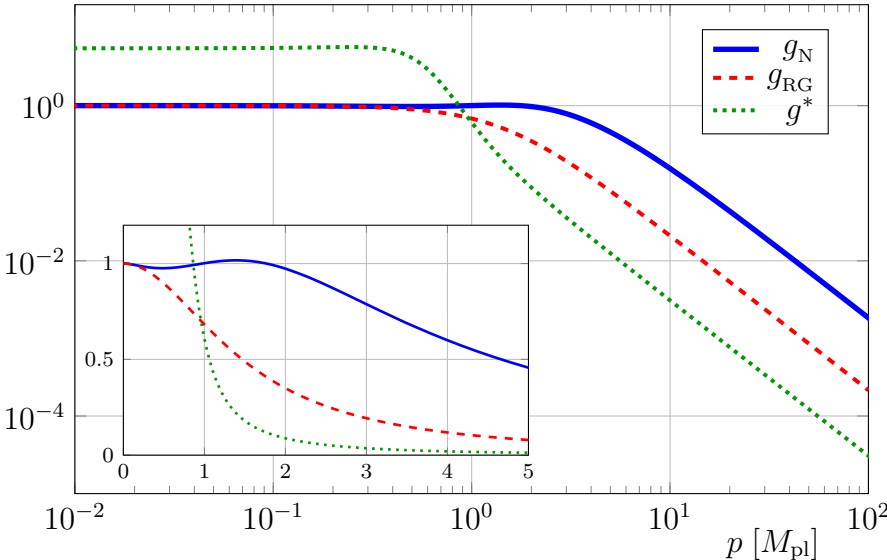

Figure 4: Physical Newton coupling $g_N(p) = G_N(p)M_{pl}^2$ as a function of momentum in units of the Planck mass (blue, solid). For comparison, we also show the scale-dependent Newton coupling $g_{RG} = g_k(p = 0)$ (red, dashed) as a function of $k = p$, and the fixed-point coupling $g^*(p/k)$ as a function of a dimensionless momentum variable (green, dotted).

### 4.2.1 Physical Newton coupling

This novel result of a momentum-dependent coupling at $k = 0$ allows us to evaluate the standard approximation of identifying the physical momentum-dependent Newton coupling at vanishing cutoff with the $k$-running of the Newton coupling at vanishing momentum, and that of the fixed-point coupling. In Fig. 4, we depicted the momentum-dependent fixed-point coupling $g^*(p/k)$, and the $k$-dependent coupling at $p = 0$, $g_{\mathrm{RG}}(k)$, for comparison:

(i) *cutoff-momentum identification* $(p-k)$: While $g_{\mathrm{RG}}(k)$ trivially agrees with the physical coupling $g_{\mathrm{N}}(p)$ for small momenta, it turns over towards the asymptotically safe fixed point running at smaller momentum scales. Indeed this happens nearly an order of magnitude earlier. Moreover, the UV coupling is also far smaller, and it does not show the intermediate rise of the coupling.

(ii) *Fixed-point identification:* The fixed-point coupling $g^*(p_{\mathrm{FP}})$, normalised with the cutoff scale $k$ lacks a determination of the (IR) Planck mass. Here, $p_{\mathrm{FP}}$ indicates that the momentum in $g^*$ is measured in the cutoff scale. This normalisation of both, the Newton coupling and the momentum, with the cutoff scale, leads to the deviation of its 'IR' value from the physical one. If rescaled to fit the IR coupling, it turns towards the asymptotically safe regime even earlier than $g_{\mathrm{RG}}(k)$. Also the UV value of the coupling is even smaller than that of $g_{\mathrm{RG}}(k)$.

In summary, we conclude that both procedures, *(i)* and *(ii)*, mimic the qualitative aspects of the physical Newton coupling. Moreover, the common $p-k$–identification, *(i)*, works considerably better than the fixed-point identification. However, the comparison also shows that both procedures cannot be used for quantitative statements. This concerns in particular physics that covers both the asymptotically safe UV regime and the classical IR regime. In both procedures, *(i)* and *(ii)*, the relative momentum scales in the two regimes are off by one or more orders of magnitude. We emphasise that while this has been shown here for the scattering coupling of three gravitons, this readily translates to other observables: the three-graviton coupling is at the root of all scattering processes.

Finally, the results of the present work can also be used to improve upon the procedures *(i)* and *(ii)* used in the literature: the comparison of $g_{\mathrm{RG}}(k)$, $g^*(p_{\mathrm{FP}})$ with the physical coupling $g_{\mathrm{N}}(p)$ allows us to establish identifications $k \to p$ and $p_{\mathrm{FP}} \to p$ for phenomenological use. Still, for more quantitative statements and scattering observables with several momentum scales this is bound to fail, and one has to resort to the full computation within the present fluctuation approach, see [14, 45–58, 68] for pure gravity and [54, 55, 57–68] for gravity-matter systems.

### 4.3 Euclidean background propagator

In this section, we relate the Newton coupling obtained in Sec. 4.2 from the fluctuation three-graviton vertex to the background graviton propagator. Similarly to the fluctuation graviton propagator in (44), the background graviton propagator is parametrised as

$$\mathcal{G}_{\bar{g}\bar{g}}(p) = \frac{1}{Z_{\bar{g}}(p)p^2}, \tag{52}$$

with the (inverse) dressing or wave-function renormalisation $Z_{\bar{g}}(p) = Z_{\bar{g},k\to0}(p)$. The latter is related with background diffeomorphism invariance to the $\beta$-function of the Newton coupling. With (50), this leads us to the relation

$$\eta_{\bar{g}}(p) = -\frac{\partial_t Z_{\bar{g}}(p)}{Z_{\bar{g}}(p)} = \eta_g(p). \tag{53}$$

In (53), we have used that we have identified all avatars of the Newton coupling with $g_k(p)$, including the background coupling. This (symmetric-point) approximation has been proven to hold true semi-quantitatively, for more details see e.g. [14, 45]. Note also that it is at the root of the background-field approximation used in the literature.

Given later applications to cross-sections and other observables, it is instructive to relate the definition of the background propagator to the tree-level scattering of fluctuation gravitons with a one-graviton exchange. Such a scattering process in the $s$-channel is displayed diagrammatically in Fig. 5. To relate it to the background propagator, we need to contract the external legs with two further fluctuation graviton propagators. This reads schematically

$$\mathcal{G}_{\bar{g}\bar{g}}(p) \simeq \mathcal{G}_{hh}(p) \big[ \Gamma^{(hhh)}(p)\mathcal{G}_{hh}(p)\Gamma^{(hhh)}(p) \big] \mathcal{G}_{hh}(p), \tag{54}$$

see also Fig. 5. Here, we have implicitly projected on transverse-traceless part of the scattering process. Note that in (54), all fluctuation wave-function renormalisations cancel out and we are left with a $g_k(p)/p^2$ behaviour in the high-momentum regime, as expected. This way of defining a background propagator or running coupling has a straightforward analogy in QCD, where the analogous tree-level process of gluon-gluon scattering can be linked to the background propagator. While not identical, they share both qualitative as well as quantitative features.

We emphasise that while (54) as well as its gluon analogue $\mathcal{G}_{\bar{A}\bar{A}}$ are reminiscent of an $s$-channel contribution to 2-to-2–scattering of particles, they do not describe an on-shell physical process: for the present case of gravity the initial and final 'states' are fluctuation gravitons, which are not diffeomorphism-invariant. For QCD the final 'states' are fluctuation gluons, which are not gauge invariant.

Note also that considering a 2-to-2–scattering of background gravitons does not improve on this situation. Despite their rôle for the construction of a diffeomorphism-invariant effective action, they are no on-shell physical particles. We recall the fact that the background gluon shares all these gauge-covariant properties with the background graviton. Still, its spectral function has negative parts.

With (53), we readily compute the Euclidean background graviton propagator or $s$-channel scattering of gravitons on the symmetric point. The result is depicted with the blue solid line in Fig. 6. The leading asymptotics of $\mathcal{G}_{\bar{g}\bar{g}}(p)$ are proportional to $1/p^2$ for small momenta, and $p^{\eta_{\bar{g}}-2} = 1/p^4$ for large momenta. Asymptotic safety requires $\eta_{\bar{g}} = -2$ at the UV fixed point, see also Sec. 3.1.

### 4.3.1 IR asymptotics

The low-momentum asymptotic of $1/p^2$ captures the classical IR-regime: the theory approaches classical gravity with the Einstein-Hilbert action in (1). This does not exclude the presence of subleading features that may carry important physics.

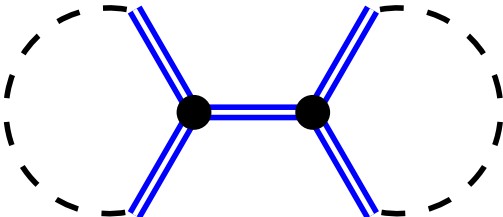

Figure 5: Tree-level graviton-graviton scattering diagram. The dashed lines indicate the transverse-traceless contraction of the external legs.

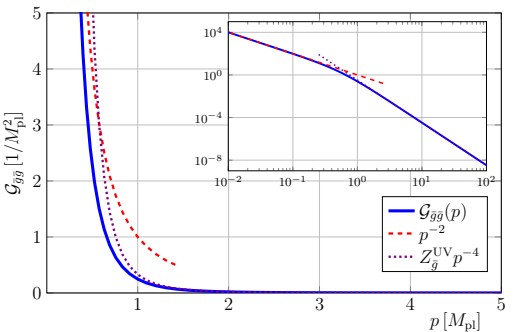 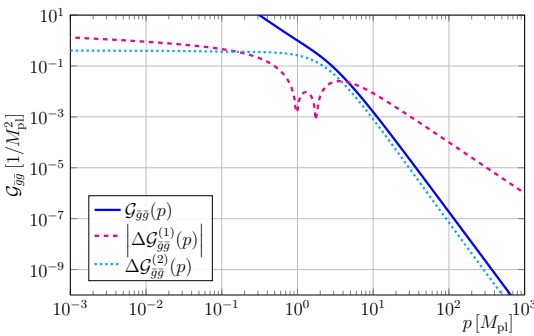

Figure 6: Momentum dependence of the Euclidean background graviton propagator $\mathcal{G}_{\bar{g}\bar{g}}(p)$. (left) Momentum dependence of the Euclidean background graviton propagator (blue, solid) with its IR and UV asymptotics. The UV asymptotic (violet, dotted) is given by $p^{-4}$ with $Z_{\bar{g}}^{\text{UV}} \approx 18.5$, see (25). The IR asymptotic (red, dashed) is simply the classical dispersion $1/p^2$. (right) Subleading contributions at small momenta in comparison to the full propagator (blue, solid). $\Delta\mathcal{G}_{\bar{g}\bar{g}}^{(1)}$ (magenta, dashed) carries a subleading log-like contribution, while $\Delta\mathcal{G}_{\bar{g}\bar{g}}^{(2)}$ (cyan, dotted) carries a constant contribution for small momenta.

To access the subleading IR behaviour, we follow the same procedure as for the fluctuation graviton propagator in Sec. 4.1.1 and subtract the $1/p^2$-pole. This leads us to the difference propagator $\Delta\hat{\mathcal{G}}_{\bar{g}\bar{g}}^{(1)}$,

$$\Delta\hat{\mathcal{G}}_{\bar{g}\bar{g}}^{(1)}(\hat{p}) = \hat{\mathcal{G}}_{\bar{g}\bar{g}}(\hat{p}) - \frac{1}{\hat{p}^2} \,. \tag{55}$$

In contradistinction to the fluctuation graviton, $\Delta\hat{\mathcal{G}}_{\bar{g}\bar{g}}^{(1)}$ is not subleading in the UV: the subtraction introduces a $1/p^2$-dependence that dominates the $1/p^4$ asymptotics. This is seen in Sec. 4.3, where the magenta dashed line depicts the resulting subleading contribution.

Similarly to the fluctuation graviton propagator, we observe a subleading log-like contribution for small momenta,

$$\lim_{\hat{p}\to 0}\Delta\hat{\mathcal{G}}_{\bar{g}\bar{g}}^{(1)}(\hat{p}) = -A_{\bar{g}}\ln\hat{p}^2 + C_{\bar{g}} \,, \tag{56}$$

with $A_{\bar{g}} = -\frac{111}{380\pi} \approx -0.093$ and $C_{\bar{g}} \approx -0.11$. The coefficient $A_{\bar{g}}$ is again computed from the derivative of the one-loop anomalous dimension, $A_{\bar{g}} = -\partial_g\partial_{p^2}\eta_{\bar{g}}(p)/2|_{g,p\to 0}$, and is regulator independent but gauge dependent, see (86) in App. F for the full gauge dependence. As in Sec. 4.1.1, we capture this contribution with the hypergeometric function $U_{1,1}$. We define

$$\Delta\hat{\mathcal{G}}_{\bar{g}\bar{g}}^{(2)}(\hat{p}) = \Delta\hat{\mathcal{G}}_{\bar{g}\bar{g}}^{(1)}(\hat{p}) - A_{\bar{g}}\,U_{1,1}(\hat{p}^2) + \frac{1+A_{\bar{g}}}{1+(\hat{p}+\Delta\Gamma_1)^2} + 2\frac{(1+A_{\bar{g}})\Delta\Gamma_1}{(1+(\hat{p}+\Delta\Gamma_2)^2)^{\frac{3}{2}}} \,. \tag{57}$$

In (57), we have employed a combination of hypergeometric functions and two BW structures, see Sec. 5.1: this parametrisation ensures that $\Delta\hat{\mathcal{G}}_{\bar{g}\bar{g}}^{(2)}(\hat{p})$ is positive for all momenta, and that its UV asymptotics is given by $1/\hat{p}^4$. The lack of sign changes in $\Delta\hat{\mathcal{G}}_{\bar{g}\bar{g}}^{(2)}$ facilitates the reconstruction, though it is not necessary. We have checked that the specific values of the $\Delta\Gamma_i$ have no impact on the reconstruction result, and the values used here are $\Delta\Gamma_1 = \Delta\Gamma_2 = 2$.

We depict $\Delta\hat{\mathcal{G}}_{\bar{g}\bar{g}}^{(2)}$ with the cyan dotted line in Sec. 4.3. It shows some smooth substructures that are related to the analytic subtractions in (57), importantly, these subtractions do not

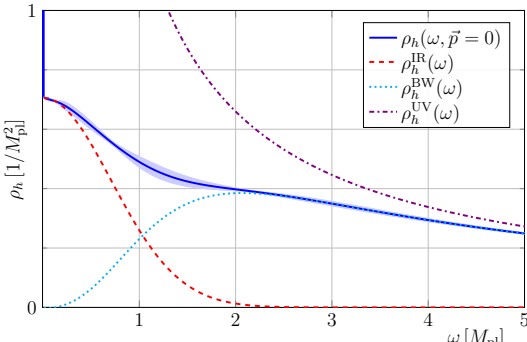
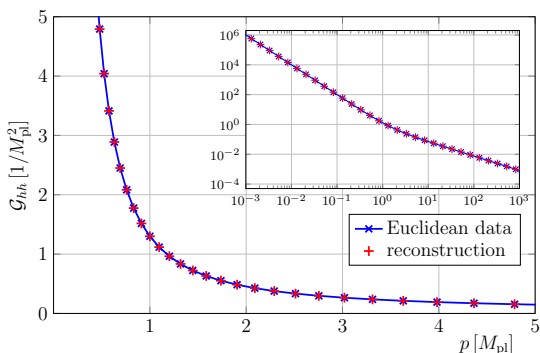

Figure 7: Spectral function of the fluctuation graviton and the reconstructed Euclidean propagator. The reconstruction, the definition of its error, and the error band in Sec. 5 are described in Sec. 5.1 and Sec. 5.2. (left) Spectral function of the fluctuation graviton (blue, solid line). It features a $\delta$-function at $\omega = 0$ (massless graviton), and an ensuing smooth multi-particle continuum, $\rho_h^{\text{cont}}(\omega)$. We also depict the analytic IR- and UV-asymptotics (red, dashed and violet, dash-dotted), and the Breit-Wigner part (cyan, dotted). (right) Euclidean fluctuation propagator reconstructed from the spectral function presented in the left panel. The reconstructed and original Euclidean data agree very well on all data points corresponding to a reconstruction error of $E_{\text{rel}} < 10^{-6}$.

introduce cuts and poles. These structures could be smoothed out, but since this does not have an impact on the resulting systematic error of the reconstruction, we refrain from doing so.

As for the fluctuation graviton, the spectral function of the asymptotic IR fits is readily computed, which leaves us only with a reconstruction task of the remaining part of the propagator, $\Delta \hat{\mathcal{G}}_{\bar{g}\bar{g}}^{(2)}$. This is done by a fit of BW structures. This is detailed in Sec. 5, the resulting spectral function is discussed in Sec. 5.3.

# 5 Graviton spectral functions

In this section, we compute the spectral functions of the fluctuation and background propagator with reconstruction methods from the Euclidean propagators computed in Sec. 4. We discuss the results for the spectral function of the fluctuation graviton $\rho_h$ in Sec. 5.2 and the one for the background graviton $\rho_{\bar{g}}$ in Sec. 5.3. These results provide an important first step towards a comprehensive understanding of the spectral properties of asymptotically safe gravity including unitarity.

## 5.1 Spectral reconstruction

As described in Sec. 4.1.1 and Sec. 4.3.1, the IR asymptotics of the Euclidean graviton propagators can be taken into account analytically in the form of a $1/\hat{p}^2$ pole and a hypergeometric function encompassing a subleading log-like pole. The remaining contributions, $\Delta \mathcal{G}^{(2)}$, are constant for small momenta and tend towards $\hat{p}^{-2+\eta}$ for large momenta. The respective anomalous dimension is dynamical for the fluctuation graviton $\eta_h \approx 1.03$, while asymptotic safety dictates $\eta_{\bar{g}} = -2$ for the background graviton.

The remaining numerical contribution $\Delta \mathcal{G}^{(2)}$ is treated with the reconstruction method described in [37], for an assessment of other reconstruction methods see also the detailed

discussion there. We proceed by choosing an ansatz of a combination of BW-like structures,

$$\hat{\mathcal{G}}^{\mathrm{BW}}(\hat{p}_0) = \mathcal{K} \sum_{k=1}^{N_{\mathrm{ps}}} \prod_{j=1}^{N_{\mathrm{pp}}^{(k)}} \left( \frac{\hat{\mathcal{N}}_k}{(\hat{p}_0 + \hat{\Gamma}_{k,j})^2 + \hat{M}_{k,j}^2} \right)^{\delta_{k,j}}. \tag{58}$$

Here, $N_{\mathrm{ps}}$ is linked to the number of BW-structures needed to describe the propagator, $N_{\mathrm{pp}}^{(k)}, \delta_{k,j}$ are linked to the shape and decay of the single structures, and $\mathcal{K}$ is an overall normalisation of the propagator, for more details see [37]. As mentioned before, we could also describe the high momentum asymptotics analytically, and only fit the remaining structures with the ansatz (58). However, since the exponents $\delta_{k,j}$ naturally lead to the same type of high momentum asymptotics, this does not improve the convergence of the reconstruction.

The fit of $\Delta\mathcal{G}^{(2)}$ with BW-like structures leads us to a fully analytical description of the propagators, see (61) for the fluctuation propagator, and (64) for the background propagator. This allows us to readily compute the spectral function $\rho$ and also to reconstruct the graviton propagator from the obtained spectral function. The reconstructed graviton propagator $\mathcal{G}^{\mathrm{rec}}$ is defined just as in (20) with

$$\mathcal{G}^{\mathrm{rec}}(p_0) = \int_0^\infty \frac{\mathrm{d}\lambda}{\pi} \frac{\lambda \, \rho(\lambda)}{\lambda^2 + p_0^2}. \tag{59}$$

The spectral function is now fixed by minimising the averaged deviation or error $E_{\mathrm{rel}}$ between the Euclidean data and its reconstruction,

$$E_{\mathrm{rel}} = \frac{1}{N} \sum_i \left( \frac{\mathcal{G}(p_i) - \mathcal{G}^{\mathrm{rec}}(p_i)}{\mathcal{G}(p_i)} \right)^2, \tag{60}$$

where the index $i$ runs over the $N$ data points considered for the fit. The relative error $E_{\mathrm{rel}}$ in (60) is measured in terms of the values $\mathcal{G}(p_i)$ of the Euclidean propagator on the data points. This definition can be further optimised as after the subtraction of the asymptotics the data points in the vicinity of the Planck scale (several orders of magnitude) are most relevant. Improved reconstructions on recently developed methods based on machine learning as well as further structural insights, see [88], will be presented elsewhere.

To minimise bias w.r.t. the choice of BW structures, we use various fits with different $N_{\mathrm{ps}}$ and $N_{\mathrm{pp}}^{(k)}$. Then we select the best fits by their relative error (60) and an additional smoothness constraint: the error defined in (60) does not punish oscillations. This introduces a well-known instability towards smaller $E_{\mathrm{rel}}$ at the expense of oscillations, for a discussion see again [37, 88] and references therein.

This finalises the set-up of our reconstruction procedure. The resulting spectral functions are discussed in the following sections Sec. 5.2 and Sec. 5.3.

## 5.2 Spectral function of the fluctuation graviton

The reconstruction method explained in Sec. 5.1 provides us with a spectral function, which is derived from the fluctuation graviton propagator,

$$\hat{\mathcal{G}}_{hh}(\hat{p}) = \frac{1}{\hat{p}^2} + A_h \, U_{1,1}(\hat{p}^2) + \hat{\mathcal{G}}_{hh}^{\mathrm{BW}}(\hat{p}), \tag{61}$$

with $\hat{\mathcal{G}}_{hh}^{\mathrm{BW}}$ defined in (58). The parameters in (61) are summarised in App. F in Tab. 1, and the resulting spectral function is shown in Sec. 5. The reconstructed propagator is in quantitative

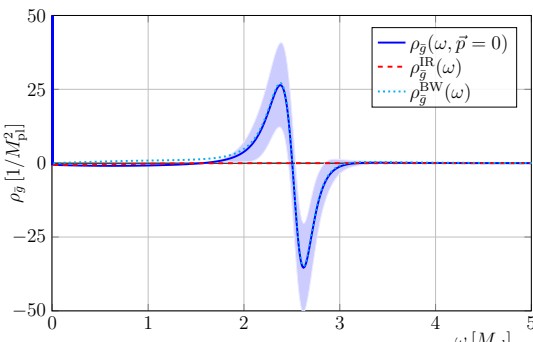
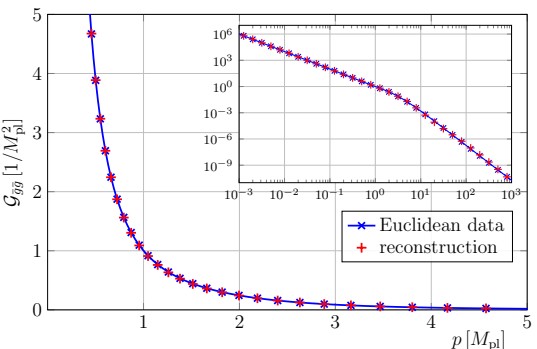

Figure 8: Spectral function of the background graviton and reconstructed Euclidean graviton propagator. The reconstruction, the definition of its error, and the error band in Sec. 5.1 are described in Sec. 5.1 and Sec. 5.3. (left) Spectral function of the background graviton (blue, solid line). It features a $\delta$-function at $\omega = 0$ (massless graviton), and an ensuing smooth multi-particle continuum, $\rho_{\bar{g}}^{\mathrm{cont}}(\omega)$. We also depict the IR asymptotics (red, dashed), and the Breit-Wigner part (cyan, dotted). (right) Euclidean background propagator reconstructed from the spectral function presented in the left panel. The reconstructed and original Euclidean data agree very well on all data points corresponding to a reconstruction error of $E_{\mathrm{rel}} < 10^{-3}$.

agreement with the Euclidean input data, see Sec. 5. The best fit for the spectral function $\rho_h$ is given by the blue, solid line, and further reconstructions within the error $E_{\mathrm{rel}} < 10^{-5}$, see (60), are indicated by the blue-shaded area. The latter provides our systematic error estimate.

We split the spectral function into two parts,

$$\hat{\rho}_h(\hat{\omega}) = \frac{\pi}{\hat{\omega}}\delta(\hat{\omega}) + \hat{\rho}_h^{\mathrm{cont}}(\hat{\omega}), \tag{62}$$

where the $\delta$-function at vanishing frequency comprises a 'classical' massless graviton and $\hat{\rho}_h^{\mathrm{cont}}(\hat{\omega})$ comprises the ensuing smooth multi-particle continuum and the UV-asymptotics. The spectral function of the fluctuation graviton shows several well-understood properties:

(i) *Classical gravity:* It has a $\delta$-function contribution at vanishing frequency due to the $1/p^2$ IR asymptotics of the Euclidean propagator. This contribution is simply that of a classical graviton propagator that arises from the curvature term in the Einstein-Hilbert action. We remind the reader in this context that we have set the cosmological constant to zero for the sake of simplicity. It can be resurrected within the computation.

(ii) *Perturbative low energy scattering spectrum:* The massless pole contribution also leads to scattering events with arbitrarily small momenta. Hence the multi-particle scattering continuum leads to a (subleading) cut mirrored in the log-like divergence of the propagator at small momenta. In terms of Cutkosky rules, these would correspond to 1-to-2 and 2-to-2 scattering events. However, we should keep in mind that the fluctuation graviton is not a physical field, and hence these are not physical scattering events. The logarithmic cut leads to a finite IR part, $\hat{\rho}_h^{\mathrm{cont}}(0) = 2\pi A_h \approx 0.71$, see Sec. 5. The scattering events from perturbative low energy gravity dominate roughly up to the Planck scale.

(iii) *IR-UV transition regime at the Planck-scale:* As expected, in the regime about the Planck scale, the BW contributions take over, and facilitate a smooth transition towards the large frequency asymptotics in the asymptotically safe UV regime. We also emphasise that this regime does not feature any pronounced structure such as an additional peak.

(iv) *Asymptotically safe regime:* The IR-UV transition in *(iii)* tends towards the UV-asymptotics in the asymptotically safe UV-regime. This asymptotics is given analytically by,

$$\lim_{\hat{\omega}\to\infty} \hat{\rho}_h^{\mathrm{UV}}(\hat{\omega}) = 2Z_h^{\mathrm{UV}} \sin\left(\eta_h \frac{\pi}{2}\right) \frac{1}{\hat{\omega}^{2-\eta_h}}, \tag{63}$$

with $\eta_h \approx 1.03$ and $Z_h^{\mathrm{UV}} \approx 0.64$. This is a direct consequence of the UV asymptotics of the Euclidean propagator, see Sec. 4.1.1. The positive value of the anomalous dimension $\eta_h$ implies that the total weight of the spectral function diverges, see the discussion in Sec. 3.1.2.

In summary, the $\delta$-function at vanishing frequency, and consequently also the scattering cut, as well as the high-frequency asymptotics, all follow directly from analytic properties of the Euclidean propagator. They do not depend on the details of the chosen reconstruction method. Another important and stable property is the positivity of the spectral function, which holds for all reconstructions. In conclusion, these are the 'physics' properties of the spectral function $\rho_h(p)$: while it is a gauge-fixed correlation function, it is, together with the Newton coupling $g_N(p)$, *the* pivotal building block of asymptotically safe gravity. In particular, the gravity contributions of scattering elements are constructed from it. Hence, the fluctuation graviton satisfies one of the necessary condition for applying Cutkosky cutting rules, see [85].

## 5.3 Spectral function of the background graviton

The reconstruction method explained in Sec. 5.1 also provides us with a spectral function, which is derived from the background graviton propagator,

$$\hat{\mathcal{G}}_{\bar{g}\bar{g}}(\hat{p}) = \frac{1}{\hat{p}^2} + A_{\bar{g}} U_{1,1}(\hat{p}^2) - \frac{1+A_{\bar{g}}}{1+(\hat{p}+\Delta\Gamma_1)^2} - \frac{2(1+A_{\bar{g}})\Delta\Gamma_1}{\left[1+(\hat{p}+\Delta\Gamma_2)^2\right]^{\frac{3}{2}}} + \hat{\mathcal{G}}_{\bar{g}\bar{g}}^{\mathrm{BW}}(\hat{p}), \tag{64}$$

with $\hat{\mathcal{G}}_{\bar{g}\bar{g}}^{\mathrm{BW}}$ defined in (58). The parameters in (64) are summarised in App. F in Tab. 2, and the resulting spectral function is shown in Sec. 5.1. The reconstructed propagator is in quantitative agreement with the Euclidean input data, see Sec. 5.1. The best fit for the spectral function $\rho_{\bar{g}}(p)$ is given by the blue, solid, line and further reconstructions within the error $E_{\mathrm{rel}} < 10^{-3}$, see (60), are indicated by the blue-shaded area. The latter provides our systematic error estimate

We again split the spectral function into two parts,

$$\hat{\rho}_{\bar{g}}(\hat{\omega}) = \frac{\pi}{\hat{\omega}}\delta(\hat{\omega}) + \hat{\rho}_{\bar{g}}^{\mathrm{cont}}(\hat{\omega}), \tag{65}$$

where the $\delta$-function at vanishing frequency comprises a 'classical' massless graviton and $\hat{\rho}_{\bar{g}}^{\mathrm{cont}}(\hat{\omega})$ comprises the ensuing smooth multi-particle continuum and the UV-asymptotics. The spectral function shows several well-understood properties:

(i) *Classical gravity:* it has a $\delta$-function contribution at vanishing frequency due to the $1/p^2$ IR asymptotics of the Euclidean propagator. This contribution is simply that of a classical graviton propagator that arises from the curvature term in the Einstein-Hilbert action. In this regime, classical diffeomorphism invariance holds up to small perturbative corrections. Hence, the normalisation of the $\delta$-function is the same as for the fluctuation graviton.

(ii) *Perturbative low energy scattering spectrum:* This massless pole contribution of the fluctuation graviton also leads to scattering events for the background graviton with arbitrarily

small momenta, and hence the multi-particle scattering continuum leads to a (subleading) cut mirrored in the log-like divergence at small momenta. However, these events differ from those in the fluctuation graviton: while the scattering events for the background graviton can be understood as tree-level (*s*-channel) 2-to-2 scatterings of gravitons, those of the fluctuation graviton are loop contributions, which may be interpreted via cutting rules also as 1-to-2 scatterings. This can lead to significant differences: $\rho_{\bar{g}}^{\text{cont}}$ has already at vanishing frequencies a negative finite value of $\hat{\rho}_{\bar{g}}^{\text{cont}}(0) = 2\pi A_{\bar{g}} = -\frac{111}{190} \approx -0.58$, see Sec. 5.1.

(iii) *IR-UV transition regime in the Planck-scale regime:* In contradistinction to the spectral function of the fluctuation graviton, that of the background graviton shows distinct peaks at frequencies about the Planck scale. In this regime, the systematic error of the reconstruction grows large. We remark that if dropping the anomalous dimension of the fluctuation graviton in the diagrams for the propagator of the background graviton, the spectral function $\rho_{\bar{g}}$ only shows the positive $\delta$-function at vanishing frequency and a negative one with the same amplitude at around the Planck scale. The anomalous dimension $\eta_h$ carries rescattering events and softens the negative $\delta$-function, leading to the pronounced peak structure in Sec. 5.1.

(iv) *Asymptotically safe regime:* The IR-UV transition in *(iii)* tends towards the UV asymptotics in the asymptotically safe UV regime. The asymptotics is given analytically by,

$$\lim_{\hat{\omega} \to \infty} \hat{\rho}_{\bar{g}}^{\text{UV}}(\hat{\omega}) = \frac{2Z_{\bar{g}}^{\text{UV}}}{\hat{\omega}^4}, \tag{66}$$

with $Z_{\bar{g}}^{\text{UV}} \approx 24.1$. Eq. (66) is dictated by asymptotic safety, see Sec. 3.1.2. It also implies that the total spectral weight vanishes, see (41). The sum of the three terms in first line on the right-hand side of (64) has already *analytically* the property (66), while the two terms in the second line individually enjoy this property. Accordingly the sum rule (41) is satisfied analytically, and we find

$$\int_0^\infty \mathrm{d}\lambda\, \lambda\, \rho_{\bar{g}}(\lambda) = 0. \tag{67}$$

Eq. (67) is analogous to the Oehme-Zimmermann super-convergence relation [82, 83] for the gluon spectral function.

In summary, the spectral function of the background graviton shows the required $\delta$-function at vanishing frequency, identical to that of the fluctuation graviton, as well as a finite low energy part that originates in perturbative scattering events. At large frequency is shows a softened negative peak that is dictated by asymptotic safety, and leads to a vanishing total spectral weight. We close this part with an important comment on the physical interpretation of background correlation functions. To that end, we use the single-graviton exchange process in Fig. 5 and its QCD analogue as a 'telling' example. In QCD, we may define a 'background' propagator $\mathcal{G}_{\bar{A}\bar{A}}(p)$ with tree-level gluon-gluon scattering or quark–anti-quark scattering: (54) and Fig. 5, where the gravitons are substituted by gluons. Then, an IR singular behaviour with $\mathcal{G}_{\bar{A}\bar{A}}(p) \propto 1/p^4$ for small momenta would give rise to confinement with a linear potential within a single gluon-exchange picture. Assuming the resulting IR dominance of gluons it has been shown that functional equations in the Landau gauge indeed admit such a solution (the Mandelstam solution) [89]. What makes this self-consistent picture even more appealing is the direct physics interpretation of the respective propagator $\mathcal{G}_{\bar{A}\bar{A}}$. However, a full computation

reveals that the propagator $\mathcal{G}_{\bar{A}\bar{A}}$ is even IR suppressed, $p^2 \mathcal{G}_{\bar{A}\bar{A}}(p) \to 0$ for small momenta [90]. Moreover, the spectral function for $\mathcal{G}_{\bar{A}\bar{A}}$ contains negative parts both for asymptotically small and large spectral values [37]: that for large spectral values is triggered by the *perturbative* tail. In turn, that for small spectral values is triggered by the ghost. Moreover, the confining potential is only revealed by an all order *gauge-invariant* computation, see [91]. In conclusion, while the direct physics interpretation of the gluon-gluon scattering is very appealing and is even confirmed within approximations based on this assumption (self-consistency), it turns out to be even qualitatively wrong. Note that this statement includes the perturbative part.

The analysis in QCD leads to an important lesson for asymptotically safe gravity: there is no evidence for the validity of a direct physics interpretation of diffeomorphism-*variant* scattering events such as graviton-graviton scatterings, in particular in the strongly correlated asymptotically safe UV regime. In turn, while such an interpretation fails in QCD even for *perturbative* UV momenta, the IR scattering of gravitons simply describes the scattering of massless particles, see Fig. 1b or Sec. 5.1.

# 6   Conclusions

In the present work, we have reported on first, but important steps towards an understanding of asymptotic safety with Lorentzian signature. In particular, we have computed the spectral functions of the fluctuation and background graviton, see Fig. 1, with reconstructions methods from the full momentum-dependent Euclidean propagators at vanishing cutoff scale. The Euclidean results also encompass a full momentum-dependent avatar of the Newton coupling at vanishing cutoff scale. In particular, they allow the discussion of scattering events as well as the benchmarking of standard phenomenological scale identifications in the literature, see Sec. 4.2.1.

The results for the spectral functions have been presented and discussed in detail in Sec. 5. For details on the fluctuation graviton we refer to Sec. 5.2, and for details on the background graviton we refer to Sec. 5.3. The spectral function of the fluctuation graviton is positive and hence the fluctuation graviton in the Landau-DeWitt gauge satisfies one of the necessary condition for applying Cutkosky cutting rules, see [85]. However, the spectral function is not normalisable, which signals the fact that the graviton is not directly related to an asymptotic state. In turn, that of the background graviton has positive and negative parts, and has a vanishing total spectral weight. Its properties are reminiscent of that of the background gluon, see the analysis at the end of the last section, Sec. 5.3. This analogy suggests in particular that the background graviton does not carry any signature of unitarity violation or preservation, which is a far more intricate matter.

Next steps include the application of the spectral functions to the computation and analysis of observables such as scattering events or the cosmological evolution induced by asymptotically safe gravity. Moreover, we currently corroborate the results obtained in the present work with a direct computation of the Minkowski propagators within the spectral setup put forward in [92]. With all these advances we aim at an investigation of unitarity in asymptotically safe gravity within the present approach.

## Acknowledgements

We thank A. Barvinsky, J. Horak, U. Moschella, and N. Wink for discussions. This work is supported by the DFG, Project-ID 273811115, SFB 1225 (ISOQUANT), as well as by the DFG under Germany's Excellence Strategy EXC - 2181/1 - 390900948 (the Heidelberg Excellence Cluster STRUCTURES). MR acknowledges funding by the Science and Technology Research Council (STFC) under the Consolidated Grant ST/T00102X/1.

## A   Gauge fixing

The gauge-fixing action is given by

$$S_{\text{gf}}[\bar{g}, h] = \frac{1}{2\alpha} \int \mathrm{d}^4 x \, \sqrt{\bar{g}} \, \bar{g}^{\mu\nu} F_\mu F_\nu, \tag{68}$$

with the de-Donder type gauge fixing condition $F_\mu$,

$$F_\mu = \bar{\nabla}^\nu h_{\mu\nu} - \frac{1+\beta}{4} \bar{\nabla}_\mu h^\nu{}_\nu. \tag{69}$$

This leads to the ghost action

$$S_{\text{gh}}[\bar{g}, h, \bar{c}, c] = \int \mathrm{d}^4 x \, \sqrt{\bar{g}} \, \bar{c}^\mu \mathcal{M}_{\mu\nu} c^\nu, \tag{70}$$

with the Faddeev-Popov operator $\mathcal{M}$ following from a diffeomorphism variation of the gauge-fixing condition,

$$\mathcal{M}_{\mu\nu} = \bar{\nabla}^\rho \left( g_{\mu\nu} \nabla_\rho + g_{\rho\nu} \nabla_\mu \right) - \frac{1+\beta}{2} \bar{g}^{\rho\sigma} \bar{\nabla}_\mu \left( g_{\nu\rho} \nabla_\sigma \right). \tag{71}$$

Throughout this work, we use $\beta = 1$ and the Landau limit $\alpha \to 0$.

## B   Regulator

We choose the regulator $R_k$ proportional to the two-point function at vanishing cosmological constant

$$R_k = \Gamma_k^{(2)}(\Lambda = 0) \cdot r(p^2/k^2). \tag{72}$$

Here, $r$ is the shape function for which we use the Litim-type cutoff function [93–95]

$$r(x) = \left( \frac{1}{x} - 1 \right) \Theta(1 - x). \tag{73}$$

## C   Transverse-traceless projection operator

We project the two- and three-point graviton correlation functions on their transverse-traceless part. The TT-projection operator is given by

$$\Pi_{\text{TT}}^{\mu\nu\rho\sigma}(p) = \frac{1}{2} \left( \Pi_{\text{T}}^{\mu\rho}(p) \Pi_{\text{T}}^{\nu\sigma}(p) + \Pi_{\text{T}}^{\mu\sigma}(p) \Pi_{\text{T}}^{\nu\rho}(p) \right) - \frac{1}{3} \Pi_{\text{T}}^{\mu\nu}(p) \Pi_{\text{T}}^{\rho\sigma}(p), \tag{74}$$

where

$$\Pi_{\text{T}}^{\mu\nu}(p) = \delta^{\mu\nu} - \frac{p^\mu p^\nu}{p^2}. \tag{75}$$

# D  Flow of the three-graviton vertex

The flow of the dimensionless three-point Newton coupling is obtained by the complete contraction of the flow of the three-graviton vertex with three TT-projection operators. We write

$$\text{Flow}_{\text{TT}}^{(hhh)} = \frac{\partial_t \Gamma_k^{(hhh)}(p_1, p_3, p_3)}{\prod_{i=1}^{3} \sqrt{G_k(p_i) Z_{h,k}(p_i)}} * \prod_{i=1}^{3} \Pi_{\text{TT}}(p_i), \tag{76}$$

where '$*$' stands for the complete contraction of the projection operators with the vertex. The denominator in (76) takes care of the wave-function renormalisations in (11). The flow (76) is evaluated at the momentum symmetric point, see (12), which makes is a function of one single momentum. This leads to flow equation for the Newton coupling of the three-point vertex as given in (50) with the anomalous dimension given by

$$\eta_g(p) = 3\eta_h(p) + G_k(0) \frac{\text{Flow}_{\text{TT}}^{(hhh)}(p) - \text{Flow}_{\text{TT}}^{(hhh)}(0)}{p^2}. \tag{77}$$

Here, we have also substituted $G_k(p) \to G_k(0)$ in front of the flow term in (77). This additional approximation weakens the decay of the flow for large momenta, and hence only influences subleading contributions.

# E  Computation of the anomalous dimension

The fluctuation graviton anomalous dimension $\eta_h(p)$ is derived from the flow of the graviton two-point function, (with $\lambda_n = 0$ and $g_{n,k} = g_{3,k}$)

$$\eta_h(p) = -\frac{32\pi}{5} \frac{\text{Flow}_{\text{TT}}^{(hh)}(p) - \text{Flow}_{\text{TT}}^{(hh)}(0)}{p^2}, \tag{78}$$

where

$$\text{Flow}_{\text{TT}}^{(hh)}(p) = \frac{\partial_t \Gamma_{k,\mu\nu\rho\sigma}^{(hh)}(p)}{Z_{h,k}(p)} \Pi_{\text{TT}}^{\mu\nu\rho\sigma}(p), \tag{79}$$

with the TT-projection operator as provided in App. C. Eq. (79) is the projection of the full fluctuation two-point flow on its TT-part. Eq. (78) is an integral equation for $\eta_h(p)$ since the momentum-loop integrals of the flow contain the anomalous dimension via the regulator insertion $\partial_t R_k$. We solve (78) in a two-step procedure: we first solve for its asymptotics, i.e., we expand for small and large $p$. Then we solve the full equation for $\eta_h(p)$ by an iterative procedure.

## E.1  Asymptotics of the anomalous dimension

The spectral reconstruction requires good understanding of the asymptotics of the Euclidean propagators, which directly relate to the asymptotics of the anomalous dimension. Here, we provide the explicit expressions of the anomalous dimension at large and small momenta.

For $p \to 0$, the anomalous dimension (78) is given by a derivative with respect to $p^2$ and we obtain

$$\eta_h(0) = -\frac{16\pi}{5} \partial_p^2 \left( \text{Flow}_{\text{TT}}^{(hh)}(p) \right) \Big|_{p \to 0} = \frac{g_k}{\pi} \left( \frac{19}{12} - \int_0^1 dq \, (3q^3 - 7q^5 + 4q^7) \, \eta_h(q) \right). \tag{80}$$

Similarly, we obtain an equation for $\eta_h(p)$ at large momenta $p > 2k$ from the definition (78),

$$
\begin{aligned}
\eta_h(p > 2k) = \frac{g_k}{\pi} \Bigg( & \frac{k^2}{p^2} \left( \frac{1}{2} + \frac{2}{3} \int_0^1 \mathrm{d}q\, q^5 (q^2 - 1)(7q^2 - 3)\, \eta_h(q) \right) \\
& + \frac{k^4}{p^4} \left( \frac{1}{4} + \frac{1}{3} \int_0^1 \mathrm{d}q\, q^7 (q^2 - 1)\, \eta_h(q) \right) \\
& + \frac{k^6}{p^6} \left( -\frac{1}{15} - \frac{1}{3} \int_0^1 \mathrm{d}q\, q^9 (q^2 - 1)\, \eta_h(q) \right) \\
& + \frac{k^8}{p^8} \left( -\frac{1}{90} - \frac{1}{15} \int_0^1 \mathrm{d}q\, q^{11} (q^2 - 1)\, \eta_h(q) \right) \Bigg).
\end{aligned}
\tag{81}
$$

Note that the series terminates after $(k/p)^8$, i.e., all higher orders are vanishing.

### E.2 Symmetrisation of the anomalous dimension

For $k \to 0$, the graviton anomalous dimension is a function of momentum-squared, $\eta_h = \eta_h(p^2)$: it can be expanded in $p^2$, i.e., no terms proportional to $(p^2)^n p$ are present, with $p = \sqrt{p^2}$. For finite cutoff-scale, $k \neq 0$, non-analyticities of the regulator function can break this property. This is well-known for the sharp cutoff as well as the Litim-type cutoff. The latter leads to contributions of the form $l_3 p^3 + l_4 p^4 + \mathcal{O}(p^5)$ to the flow of the graviton two-point function, while the former would even introduce terms linear in $p$. This entails that for the Litim-type cutoff, $\eta_h(p)$ as defined in (78), contain terms linear in $p$, which have to integrate to zero for $k \to 0$. This implies a fine-tuning condition for the anomalous dimension at the large initial cutoff scale $k/M_{\mathrm{pl}} \to \infty$. This fine-tuning is straightforward but tedious.

Since these terms are subleading, we refrain from the fine-tuning and only consider the even part of the anomalous dimension at each cutoff step. This is done via a Padé fit,

$$
\eta_h^{\mathrm{sym}}(p) = \frac{\eta_h^{\mathrm{Padé}}(p) + \eta_h^{\mathrm{Padé}}(-p)}{2}.
\tag{82}
$$

The Padé fit, or any other fit, introduces an analytic bias for our reconstruction. Note however that this bias only is an issue for small momenta, and not for the large momentum regime we are mainly interested in. Moreover, the flows do allow for Taylor expansions in the asymptotic regimes, which further stabilises Padé approximants.

## F Parameters of the spectral functions

In this appendix, we provide the parameters of the BW fits used for construction of the fluctuation and background spectral functions presented in Sec. 5.

Table 1: Parameters for the reconstruction of the fluctuation graviton spectral function (central line in Sec. 5).

| parameter | $\hat{\mathcal{G}}_{hh}^{\mathrm{BW}(1)}$ | $\hat{\mathcal{G}}_{hh}^{\mathrm{BW}(2)}$ | $\hat{\mathcal{G}}_{hh}^{\mathrm{BW}(3)}$ | $\hat{\mathcal{G}}_{hh}^{\mathrm{BW}(4)}$ |
|---|---|---|---|---|
| $\mathcal{K}$ | 1 | 1 | 1 | 1 |
| $\hat{\mathcal{N}}_1$ | 0.517 | 0.408 | 0.552 | 0.411 |
| $\hat{\Gamma}_{1,1}$ | 0 | 0.222 | 1.20 | 1.05 |
| $\hat{M}^2_{1,1}$ | $2.28\cdot10^{-5}$ | $1.57\cdot10^{-3}$ | 0.237 | 0.298 |
| $\delta_{1,1}$ | $-0.380$ | $-1.52$ | 2.31 | 1.34 |
| $\hat{\Gamma}_{1,2}$ | 1.43 | 0.196 | 0.628 | 0.722 |
| $\hat{M}^2_{1,2}$ | 0.562 | 0.0577 | 1.68 | $4.89\cdot10^{-7}$ |
| $\delta_{1,2}$ | 1.06 | 1.36 | 1.25 | $-0.859$ |
| $\hat{\Gamma}_{1,3}$ | 1.48 | 1.29 | | |
| $\hat{M}^2_{1,3}$ | 1.30 | 0.783 | | |
| $\delta_{1,3}$ | 0.670 | 0.647 | | |
| $\hat{\mathcal{N}}_2$ | 0.409 | 0.0649 | 0.410 | 0.260 |
| $\hat{\Gamma}_{2,1}$ | 1.75 | 1.15 | 1.30 | 0.441 |
| $\hat{M}^2_{2,1}$ | 0.546 | 1.15 | 0.657 | 2.60 |
| $\delta_{2,1}$ | 0.0133 | 1.57 | 0.632 | 1.59 |
| $\hat{\Gamma}_{2,2}$ | 1.63 | 0.389 | 0.298 | 1.27 |
| $\hat{M}^2_{2,2}$ | 1.64 | 1.17 | 0 | 1.98 |
| $\delta_{2,2}$ | 0.470 | 0.479 | $-0.149$ | 1.47 |
| $\hat{\Gamma}_{2,3}$ | | 1.37 | | 1.06 |
| $\hat{M}^2_{2,3}$ | | 0.487 | | 1.18 |
| $\delta_{2,3}$ | | 1.18 | | 1.39 |
| $\hat{\mathcal{N}}_3$ | 0.335 | $1.03\cdot10^{-3}$ | 0.0292 | $2.38\cdot10^{-5}$ |
| $\hat{\Gamma}_{3,1}$ | 0.615 | 2.48 | 1.90 | 2.59 |
| $\hat{M}^2_{3,1}$ | 0.710 | 1.16 | 1.11 | 0.641 |
| $\delta_{3,1}$ | 1.42 | 2.06 | 0.940 | 2.38 |
| $\hat{\Gamma}_{3,2}$ | 1.82 | 1.18 | 1.55 | 1.77 |
| $\hat{M}^2_{3,2}$ | 1.50 | 1.08 | 1.08 | 1.15 |
| $\delta_{3,2}$ | 2.21 | 0.618 | 0.758 | 1.31 |
| $\hat{\Gamma}_{3,3}$ | 0.329 | 2.20 | | |
| $\hat{M}^2_{3,3}$ | 1.22 | 0.991 | | |
| $\delta_{3,3}$ | 1.28 | 1.64 | | |
| $E_{\mathrm{rel}}$ | $6.67\cdot10^{-7}$ | $6.82\cdot10^{-7}$ | $1.47\cdot10^{-6}$ | $2.71\cdot10^{-6}$ |

Table 2: Parameters for the reconstruction of the background graviton spectral function (central line in Sec. 5.1).

| parameter | $\hat{\mathcal{G}}_{\bar{g}\bar{g}}^{\mathrm{BW}(1)}$ | $\hat{\mathcal{G}}_{\bar{g}\bar{g}}^{\mathrm{BW}(2)}$ | $\hat{\mathcal{G}}_{\bar{g}\bar{g}}^{\mathrm{BW}(3)}$ |
|---|---|---|---|
| $\mathcal{K}$ | 1 | 1 | 1 |
| $\hat{\mathcal{N}}_1$ | 4.25 | 4.32 | 2.81 |
| $\hat{\Gamma}_{1,1}$ | 0.190 | 0.210 | 2.41 |
| $\hat{M}_{1,1}^2$ | 2.55 | 2.56 | 3.49 |
| $\delta_{1,1}$ | 2.24 | 2.24 | $-0.637$ |
| $\hat{\Gamma}_{1,2}$ | | | 0.261 |
| $\hat{M}_{1,2}^2$ | | | 2.49 |
| $\delta_{1,2}$ | | | 2.65 |
| $\hat{\mathcal{N}}_2$ | 2.02 | 2.04 | |
| $\hat{\Gamma}_{2,1}$ | 4.01 | 4.45 | |
| $\hat{M}_{2,1}^2$ | 2.58 | 1.96 | |
| $\delta_{2,1}$ | 1.97 | 1.98 | |
| $E_{\mathrm{rel}}$ | $4.68 \cdot 10^{-4}$ | $4.78 \cdot 10^{-4}$ | $7.34 \cdot 10^{-4}$ |

## F.1 Fluctuation spectral function

We describe the Euclidean fluctuation propagator, used to reconstruct the spectral function shown in Sec. 5, by a combination of a $p^{-2}$ peak, a hypergeometric function, and the average of four BW fits to the remaining structures,

$$\hat{\mathcal{G}}_{hh}(\hat{p}) = \frac{1}{\hat{p}^2} + A_h\, U_{1,1}(\hat{p}^2) + \frac{1}{4}\sum_{i=1}^{4} \hat{\mathcal{G}}_{hh}^{\mathrm{BW}(i)}(\hat{p}), \tag{83a}$$

with

$$A_h = \frac{7}{20\pi} \approx 0.11. \tag{83b}$$

The four BW fits used were selected from a range of BW structures with different $N_{\mathrm{ps}}$ and $N_{\mathrm{pp}}^{(k)}$, c.f. (58), and all exhibit relative errors $E_{\mathrm{rel}} < 10^{-5}$. We present their fit parameters in Tab. 1.

## F.2 Background spectral function

For the reconstruction of the background spectral function shown in Sec. 5.1, we introduced two additional terms facilitating an easier fit of the remaining structures,

$$\hat{\mathcal{G}}_{\bar{g}\bar{g}}(\hat{p}) = \frac{1}{\hat{p}^2} + A_{\bar{g}}\, U_{1,1}(\hat{p}^2) - \frac{1+A_{\bar{g}}}{1+(\hat{p}+\Delta\Gamma_1)^2} - \frac{2(1+A_{\bar{g}})\Delta\Gamma_1}{\left[1+(\hat{p}+\Delta\Gamma_2)^2\right]^{\frac{3}{2}}} + \frac{1}{3}\sum_{i=1}^{3} \hat{\mathcal{G}}_{\bar{g}\bar{g}}^{\mathrm{BW}(i)}(\hat{p}), \tag{84a}$$

with

$$A_{\bar{g}} = -\frac{111}{380\pi} \approx -0.09. \tag{84b}$$

The three fits were chosen as the best fits with $E_{\mathrm{rel}} < 10^{-3}$, and we show their parameters in Tab. 2.

### F.3 Gauge dependence of $A_h$ and $A_{\bar{g}}$

The coefficients $A_h$ and $A_{\bar{g}}$ describe the logarithmic branch cuts of the fluctuation and background propagator, respectively. Therefore, they relate to a $p^4$ derivative of the flow and are independent of the choice of regulator, (72). The coefficients still depend on the gauge-fixing parameters $\alpha$ and $\beta$ as defined in (68) and (69). The full gauge dependent result of $A_h$ and $A_{\bar{g}}$ is given by

$$
\begin{aligned}
A_h = {}& \alpha^2 \frac{3\beta^4 - 36\beta^3 + 162\beta^2 - 324\beta + 259}{12\pi(\beta - 3)^4} + \alpha \frac{9\beta^4 - 90\beta^3 + 320\beta^2 - 630\beta + 519}{24\pi(\beta - 3)^4} \\
& - \frac{43\beta^4 - 406\beta^3 + 312\beta^2 + 1926\beta - 2547}{120\pi(\beta - 3)^4},
\end{aligned}
\tag{85}
$$

$$
\begin{aligned}
A_{\bar{g}} = {}& \alpha^2 \frac{-107\beta^4 + 1284\beta^3 - 5778\beta^2 + 11556\beta - 9771}{228\pi(\beta - 3)^4} \\
& + \alpha \frac{-3637\beta^4 + 37850\beta^3 - 137920\beta^2 + 233182\beta - 121155}{2280\pi(\beta - 3)^4} \\
& + \frac{7863\beta^4 - 81316\beta^3 + 272746\beta^2 - 390084\beta + 180135}{2280\pi(\beta - 3)^4}.
\end{aligned}
\tag{86}
$$

For the values $\alpha = 0$ and $\beta = 1$, these expressions reduce to (83b) and (84b) in agreement with [86]. The full gauge dependence of (85) and (86) does, however, not agree with [86]. The cause of this disagreement remains to be investigated.

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
