# Peer review of "Reconstructing the graviton"

_SciPost Physics, doi:SciPost Phys. 12, 001 (2022)_

## Round 1 · Referee Report · Anonymous (Referee 1) · 2021-3-19

Strengths

1 - very original, milestone for asymptotically safe quantum gravity
2 - very good balance between readability and technical detail

Weaknesses

1 - some of the (technical) assumptions could be better motivated/explained

Report

The paper discusses an important topic in quantum gravity: the reconstruction of a Minkowski spectral function from Euclidean data. This is a milestone on the route to obtaining results valid in Lorentzian signature. As such, it easily passes the journal's criteria for originality, excellence and significance. The authors carefully discuss the general reconstruction procedure, and then carry it out for results obtained in both the background field approximation as well as a dynamical fluctuation computation in the context of Asymptotic Safety.

The main results are that the background spectral function is not positive, and integrates to zero, while the fluctuation spectral function is positive but not normalisable. Some of the implications of this result are discussed.
As a further contribution, the authors discuss different ways to approximate the momentum dependence of Newton's constant: based on the fixed point condition, an identification of the RG scale and the momentum running, and the fully momentum dependent computation.
On the technical side, the authors are the first to present results on the momentum dependence in Asymptotic Safety at vanishing IR cutoff. This allows them to connect more directly to physical observables, without potential artefacts induced by a finite cutoff scale.

Having said all that, there are a few points that I would like the authors to consider before I can fully recommend publication of the article. These are:

1 - The authors mention several times that the graviton is not an on-shell physical field. In my opinion, this needs a little more discussion. In particular, what is the concrete difference to a photon, and what is the connection of the graviton to (linearised) gravitational waves, which have been recently observed?

2 - I have some concerns regarding the regulator choice. The authors use the "Litim cutoff", which is not smooth, to obtain momentum-dependent results. Can one be sure that this non-smoothness doesn't percolate into the reconstruction of the spectral function? And what is the actual gain of this choice, since results have to be obtained numerically anyway?

3 - To me it seems that the authors make one additional, but unstated, assumption when they relate the propagator and the spectral function. Namely, pairs of complex conjugate poles play a distinguished role. They don't contribute to the spectral function, so that the reconstruction of the propagator via the spectral function is incomplete, and these poles have to be re-added by hand (see e.g. [Phys.Rev.D 99 (2019) 7, 074001] for a discussion in the context of Yang-Mills theory). If such CC poles are present, this modifies the discussion of the analytic structure of the spectral function. While a more complete discussion would obviously be desirable, I would suggest to at least explicitly state this assumption.

4 - On the comparison between background and fluctuation momentum dependence, my impression is that this is not really an apples-to-apples comparison. Concretely, the fluctuation computation is performed with the full physical momentum dependence. By contrast, at the background level, the authors only discuss the momentum dependence obtained by identifying the RG scale running with the physical momentum dependence. A more complete computation would obtain this momentum dependence by resolving form factors quadratic in curvature. This also opens the possibility to get a different fall-off for the background propagator. This should be stated more clearly.

A list of minor comments/suggestions can be found below.

Requested changes

Section I:

1 - typo: "is not a on-shell field" -> "is not an on-shell field"

Section II:

2 - The authors mention a lack of numerically accessible Lorentzian formulations, but strictly speaking, CDT would qualify as a non-perturbative lattice formulation with a well-defined Wick rotation

3 - As an optional suggestion, the authors might want to consider adding some literature on the topic of Wick rotations in the context of gravity, e.g. [Class.Quant.Grav. 36 (2019) 10, 105008]

Subsection A:

4 - The authors state that in the IR, the action should reduce to GR. I find this a little bit misleading, since there are e.g. EFT corrections in the form of the well-known one-loop logarithms. Also, more non-local structures have been discussed in the literature.

Subsection B:

5 - As an optional suggestion, one could mention that in general the wave function renormalisation is tensorial, and that the authors choose it to be proportional to the identity in their approximation.

6 - It would be helpful if the authors would state the initial condition for the integration of the wave function renormalisation explicitly here.

Section III:

7 - typo: "the classical spectral is ultralocal" -> "the classical spectral function is ultralocal"

8 - It would be helpful if the term ultralocal mentioned in point 7 would be defined.

9 - typo: "can readily performed" -> "can readily be performed"

Subsection A:

10 - Eq. (24) excludes more general behaviour like in nonlocal gravity theories (propagator with exponential fall-off) - why is this behaviour excluded here?

11 - In my opinion it would be helpful to define the meaning of $\xi$ a bit earlier, around eq. (36).

12 - The discussion of the case $\eta<-2$ is not clear enough, and a more explicit discussion would be helpful. In particular, the divergence for small frequencies cannot be read off from the large frequency behaviour.

13 - On a more general ground, how much does the discussion rely on the concept of momentum locality introduced in [Phys.Rev.D 92 (2015) 12, 121501] by some of the present authors and others? As far as I understand, the relation between the anomalous dimension at vanishing momentum and the fall-off of the propagator and large momentum rely on momentum locality.

Section IV:

Subsection A:

14 - Eq. (45): I would suggest to change the boundaries of the integral: $-\int_k^\Lambda$ instead of $\int_\Lambda^k$

15 - It seems to me that the coefficient $A_h$ should be related to the prefactor of the one-loop universal logarithm from EFT. Is a direct comparison possible? If not, why?

16 - What is the concrete motivation to choosing the hypergeometric function $U_{a,b}$?

Subsection C:

17 - Eq. (54) needs a little more explanation. In particular, what is the index structure, and where exactly does this relation come from?

Section V:

Subsection C:

18 - typo: "soften negative peak" -> "softened negative peak"

19 - Is there an explanation for why the reconstruction of the background spectral function is so much less stable than the reconstruction of the fluctuation spectral function?

Appendix E:

20 - Eq. (E4): I think that one has a strict equality in this equation for the chosen regulator and $p^2 \geq 4k^2$. In this regime, the regulator depending on the sum of loop and external momentum vanishes identically, and the asymptotic formula should be exact.

  • validity: high
  • significance: top
  • originality: top
  • clarity: top
  • formatting: excellent
  • grammar: excellent

Author:  Manuel Reichert  on 2021-07-09  [id 1560]

(in reply to Report 1 on 2021-03-19)

We thank the referee for their careful work, their detailed comments, and also for offering helpful suggestions for improvement. In the following, we cite the comments of the referee and reply subsequently.

"1 - The authors mention several times that the graviton is not an on-shell physical field. In my opinion, this needs a little more discussion. In particular, what is the concrete difference to a photon, and what is the connection of the graviton to (linearised) gravitational waves, which have been recently observed?"

Answer: The observed gravitons at LIGO are included in the delta peak of the computed spectral function, which directly relates to the classical behaviour of the massless graviton. However, gravitational waves are not asymptotic states, the amplitude of gravitational waves decays according to Price's law, $1/v^p$, where $v$ is the advanced time and $p\geq 12$ (this is a consequence of the no-hair theorem). There is obviously no contradiction to LIGO observation since the observation takes place at finite time and distance from the source. A further important difference to the photon is linked to the fact that the gauge group of QED is linear.

"2 - I have some concerns regarding the regulator choice. The authors use the "Litim cutoff", which is not smooth, to obtain momentum-dependent results. Can one be sure that this non-smoothness doesn't percolate into the reconstruction of the spectral function? And what is the actual gain of this choice, since results have to be obtained numerically anyway?"

Answer: The non-analyticity of the Litim cutoff at finite RG scale leads to odd powers of momenta in the anomalous dimension, which are absent at vanishing RG scale. How this is ensured is discussed in App. E2. We emphasise that the choice of the Litim-type cutoff is not necessary for the computation. We have used it to connect to our previous works and it also offers the benefit that one has analytic expressions for large and small momenta.

"3 - To me it seems that the authors make one additional, but unstated, assumption when they relate the propagator and the spectral function. Namely, pairs of complex conjugate poles play a distinguished role. They don't contribute to the spectral function, so that the reconstruction of the propagator via the spectral function is incomplete, and these poles have to be re-added by hand (see e.g. [Phys.Rev.D 99 (2019) 7, 074001] for a discussion in the context of Yang-Mills theory). If such CC poles are present, this modifies the discussion of the analytic structure of the spectral function. While a more complete discussion would obviously be desirable, I would suggest to at least explicitly state this assumption."

Answer: Indeed, complex conjugated poles are absent in a standard Källén–Lehmann spectral representation. They may signal the loss of unitarity and a respective discussion has now been included. We also have added references where issues of complex conjugated poles in the context of QCD are discussed.

"4 - On the comparison between background and fluctuation momentum dependence, my impression is that this is not really an apples-to-apples comparison. Concretely, the fluctuation computation is performed with the full physical momentum dependence. By contrast, at the background level, the authors only discuss the momentum dependence obtained by identifying the RG scale running with the physical momentum dependence. A more complete computation would obtain this momentum dependence by resolving form factors quadratic in curvature. This also opens the possibility to get a different fall-off for the background propagator. This should be stated more clearly."

Answer: Contrary to the statement of the referee, the physical momentum dependence of the background propagator is computed at vanishing RG scale and we do not use an identification of the RG scale with physical momenta. We compute the physical momentum dependence of the three-graviton vertex and use background diffeomorphism invariance to map this information to the background propagator.

We also thank the referee for the minor comments/suggestions. We have modified our manuscript accordingly. To some of the comments, we reply below.

"2 - The authors mention a lack of numerically accessible Lorentzian formulations, but strictly speaking, CDT would qualify as a non-perturbative lattice formulation with a well-defined Wick rotation"

Answer: We agree with the referee that CDT has a well-defined Wick rotation. Nonetheless, there are no non-perturbative computations directly in Lorentzian signature.

"4 - The authors state that in the IR, the action should reduce to GR. I find this a little bit misleading, since there are e.g. EFT corrections in the form of the well-known one-loop logarithms. Also, more non-local structures have been discussed in the literature."

Answer: The large-scale physics in the IR is well described by the Einstein-Hilbert action and logarithmic corrections are subleading. We clarified the formulation in the manuscript.

"10 - Eq. (24) excludes more general behaviour like in nonlocal gravity theories (propagator with exponential fall-off) - why is this behaviour excluded here?"

Answer: We indeed did not include an exponential fall-off behaviour in Eq. (24), which can arise in non-local gravity theories, since those are not known to appear in asymptotically safe quantum gravity and they are not appearing in our approximation either. We included this information in the manuscript.

"12 - The discussion of the case $\eta <−2$ is not clear enough, and a more explicit discussion would be helpful. In particular, the divergence for small frequencies cannot be read off from the large frequency behaviour."

Answer: For $\eta <−2$, the propagator of the fixed-point theory is not plane-wave normalisable anymore, but indeed the propagator could be plane-wave normalisable on trajectories that run out of the fixed point. However, we do not consider such a QFT. We modified the respective paragraph to make this issue clearer.

"13 - On a more general ground, how much does the discussion rely on the concept of momentum locality introduced in [Phys.Rev.D 92 (2015) 12, 121501] by some of the present authors and others? As far as I understand, the relation between the anomalous dimension at vanishing momentum and the fall-off of the propagator and large momentum rely on momentum locality."

Answer: Our discussion in Sec. 3 does not rely on momentum locality as we consider the fall-off behaviour of the propagator at vanishing cutoff and large momenta. For theories with momentum locality, such as the one at hand, this fall-off behaviour is directly related to the anomalous dimension at $k = \infty$ and $p = 0$. For theories without momentum locality, the situation is more intricate and goes beyond the scope of this paper.

"15 - It seems to me that the coefficient Ah should be related to the prefactor of the one-loop universal logarithm from EFT. Is a direct comparison possible? If not, why?"

Answer: We thank the referee for this remark. The coefficients $A_h$ and $A_{\bar g}$ are indeed regulator independent, though gauge dependent. We included now the gauge-dependent result in our manuscript and compare it to the EFT results.

"16 - What is the concrete motivation to choosing the hypergeometric function $U_{a,b}$?"

Answer: This choice was motivated by the fact that the hypergeometric function in question describes the log-like contributions we observe at small momenta, while simultaneously neither interfering with the UV behaviour nor the reconstruction. We have added some comments explaining this choice in more detail in the manuscript.

"17 - Eq. (54) needs a little more explanation. In particular, what is the index structure, and where exactly does this relation come from?"

Answer: We use a specific channel of the graviton-graviton scattering amplitude. We have added a few more details in the description, and hope that the relation and the tensor structure are clear now.

"19 - Is there an explanation for why the reconstruction of the background spectral function is so much less stable than the reconstruction of the fluctuation spectral function?"

Answer: We have less numerical accuracy of the Euclidean data in the background case compared to the fluctuation one. In addition, the background propagator decays faster in the UV, which makes the reconstruction more challenging. In our opinion, these are the main sources for the reduced stability of the reconstruction of the background propagator.

---

## Round 1 · Referee Report · Anonymous (Referee 2) · 2021-3-28

Strengths

1-Original interesting paper

Weaknesses

1-What is the crucial point to make the Euclidean results to Lorentzian spectral function is not clearly explained, but this is the most important point of this paper.

Report

This paper tries to reconstruct the spectral function of the graviton propagator in Lorentzian signature. They first review the case for Euclidean case, and use the results already derived in their previous paper to lead to Lorentzian case. While the aim is of interest, the presentation in sect. 5, which gives the main result for Lorentzian case, was not very clear. They refer heavily to their earlier paper and the explanation is not self-contained. It is not even clear what is the crucial point to make the Euclidean result to Lorentzian and what is the main difference. This is particularly important in the present case because the obtained results for Lorentzian are quite similar to Euclidean case. I would like to ask the authors to make clear these points.

A minor remark. They derive asymptotic behavior in (31), but I think that there is inconsistency in the exponent with (29), and if this is true I seem to disagree with their claim of asymptotic behavior p^{-3} below eq.(31). They should carefully check this.

Requested changes

As described above.

  • validity: good
  • significance: good
  • originality: good
  • clarity: low
  • formatting: excellent
  • grammar: excellent

Author:  Manuel Reichert  on 2021-07-09  [id 1561]

(in reply to Report 2 on 2021-03-28)

We thank the referee for their careful work, their detailed comments, and also for offering helpful suggestions for improvement. In the following, we cite the comments of the referee and reply subsequently.

"This paper tries to reconstruct the spectral function of the graviton propagator in Lorentzian signature. They first review the case for Euclidean case, and use the results already derived in their previous paper to lead to Lorentzian case. While the aim is of interest, the presentation in sect. 5, which gives the main result for Lorentzian case, was not very clear. They refer heavily to their earlier paper and the explanation is not self-contained. It is not even clear what is the crucial point to make the Euclidean result to Lorentzian and what is the main difference. This is particularly important in the present case because the obtained results for Lorentzian are quite similar to Euclidean case. I would like to ask the authors to make clear these points."

Answer: The main point of our paper is the reconstruction of the graviton spectral function. We achieved this goal as this is the first reconstruction in a non-perturbative setup. We have not displayed the graviton propagator on the Lorentzian axis as this information is contained in the spectral function. We tried to improve the presentation in Sec. 5 and hope that the section is now more accesible to the reader.

"They derive asymptotic behavior in (31), but I think that there is inconsistency in the exponent with (29), and if this is true I seem to disagree with their claim of asymptotic behavior $p^{-3}$ below eq.(31). They should carefully check this."

Answer: Eq. (31) is describing the fall-off behaviour for the anomalous dimension of the background graviton and is only valid for $-2<\eta < 0$. We now also explicitly state the fall-off behaviour for other values of the anomalous dimension, $0 \leq \eta < 2$.

---

## Round 2 · Referee Report · Anonymous (Referee 1) · 2021-8-9

Report

The authors have answered all my concerns satisfactorily. There is however one new concern that I would like to bring forward. Comparing to the first version, the reconstruction of the background spectral function seems to have changed in the new version, but no changes have been indicated in the authors' reply/comments on resubmission. This in particular concerns the prefactor of the logarithm/the hypergeometric function ($A_{\bar g}$) related to the parameterisation of the propagator, but also the qualitative behaviour of the spectral function itself. I noticed that the fit parameters $\Delta\Gamma_{1,2}$ have changed from 2 to 5, but the authors write that their values have no impact on the reconstruction. Table II has also changed considerably. The background propagator itself shows no noticeable difference. The authors should explain what has happened here, and how this is in agreement with the error estimates for the reconstructed spectral function.

If the above concern is addressed adequately, I would be happy to recommend the paper for publication.

As a final minor point, following up on the question of numerically accessible Lorentzian formulations, it seems that there are some new developments in the field of spin foams, see [2104.00485]. I leave it to the authors whether they want to include a reference to this.

  • validity: -
  • significance: -
  • originality: -
  • clarity: -
  • formatting: -
  • grammar: -

Author:  Manuel Reichert  on 2021-08-23  [id 1699]

(in reply to Report 1 on 2021-08-09)

We apologise for the confusion regarding the computation of the background spectral function. In the previous version of this work, we had treated the low momentum dependence only numerically, which lead to an oversight of a contribution to the log-like divergence. In the present version, we computed the log-like IR contributions analytically (c.f. the computation of $A_{\bar g}$), which allowed us to improve the numerics in the IR. This has resulted in a slightly simpler background spectral function (a smaller number of Breit-Wigner structures in Table 2) and a smaller error in the reconstruction ($E_\text{rel} < 10^{-3}$ vs $E_\text{rel} < 10^{-2}$ before). The only remarkable change in the background spectral function is that it now starts negative in the deep IR, otherwise all features remained qualitatively the same. The different choice of parameters $\Delta\Gamma_{1,2}$ has no impact on the reconstruction.

We hope that this clears up the referee's concern. We also thank the referee for bringing the reference to our attention, which we will cite in the next version of our work.

---

## Round 2 · Referee Report · Anonymous (Referee 2) · 2021-8-25

Report

Though the authors said that they tried to improve the presentation in sect. 5, I do not see much change. However, there are several other places that they improved. The fact that the spectral function is the object that is important, is emphasized. About the second point I made, the author gave additional explanation and this is fine. There are several errors in the manuscript, like reference to Sec.IV C above eq.(55) which is inside Sec.IV C itself, and I suppose they mean that Sec.IV A. I recommend the authors to take another careful check to remove this kind of minor mistakes. After that, I think that this paper may be published.

  • validity: -
  • significance: -
  • originality: -
  • clarity: -
  • formatting: -
  • grammar: -

Author:  Manuel Reichert  on 2021-10-06  [id 1815]

(in reply to Report 2 on 2021-08-25)

We thank the referee for pointing out the reference error, which we have corrected. We carefully checked the manuscript for further minor mistakes.

---

## Round 3 · List of Changes

Updated references, corrected typos

---

## Editorial Decision

published